

Analysis of 24 years of mesopause region OH rotational temperature
observations at Davis, Antarctica. Part 2: Evidence of a quasi-
quadrennial oscillation (QQO) in the polar mesosphere.
W. John R. French[1], Andrew R. Klekociuk[1,2] and Frank J. Mulligan[3]
[1]Australian Antarctic Division, 203 Channel Hwy, Kingston, Tasmania, 7050, Australia
[2]Department of Physics, University of Adelaide, Adelaide, 5005, Australia
[3]Maynooth University, Maynooth, Co. Kildare, Ireland
*Correspondence to:* W. John R. French (john.french@aad.gov.au)



## Abstract

Observational evidence of a quasi-quadrennial oscillation (QQO) in the polar mesosphere is presented based on the analysis of 24 years of hydroxyl (OH) nightglow rotational temperatures derived from scanning spectrometer observations above Davis Research Station, Antarctica (68°S, 78°E). After removal of long term trend and solar cycle responses, the residual winter mean temperature variability contains an oscillation over an approximately 3.5 - 4.5 year cycle with an amplitude of 3 - 4 K. Here we investigate this QQO feature in the context of the global temperature, pressure, wind and surface fields using the Aura/MLS and TIMED/SABER satellite data, ERA5 reanalysis and the Extended-Reconstructed Sea Surface Temperature and Optimally-Interpolated sea ice concentration data sets. We find a significant anti-correlation between the QQO and the meridional wind at 86 km altitude measured by a medium frequency spaced antenna radar at Davis. The QQO signal is also correlated with vertical transport as determined from evaluation of carbon monoxide (CO) concentrations in the mesosphere. Together this relationship suggesting that a substantial part of the QQO is the result of adiabatic heating and cooling driven by the meridional flow. The presence of quasi-stationary or persistent patterns in the ERA5 data geopotential anomaly and the meridional wind anomaly data during warm and cold phases of the QQO suggests a tidal or planetary wave influence in its formation, which may act on the filtering of gravity waves to drive an adiabatic response in the mesosphere. The QQO signal potentially arises from an ocean-atmosphere response, and appears to have a signature in Antarctic sea ice extent.





# 1. Introduction

In Part 1 of this study (French et al., 2019) we quantify the solar cycle and long term trend in 24 years of hydroxyl (OH) rotational temperature measurements from Davis Research Station, Antarctica and observed that the winter mean residual temperatures revealed a periodic oscillation over an approximately 4 year cycle with amplitude of 3-4 K. While periodic oscillations occur on many timescales in the atmosphere from minutes to years (gravity waves, tides, planetary waves, seasonal variations, quasi-biennial oscillation (QBO), El Nino Southern Oscillation (ENSO), Pacific Decadal Oscillation (PDO), solar cycle), the 4-year period of this quasi-quadrennial oscillation (QQO) is unusual in terms of weather and climate modes. Here we seek to characterize the features and extent of the observed behavior and to examine correlation and composites with several atmospheric parameters which might suggest a possible mechanism for the phenomenon.

References to quasi-quadrennial variability in Earth's climate system have previously been reported by Jiang et al. (1995) who found both quasi-quadrennial (52 month) and quasi-biennial (24 and 28 months) oscillation modes in equatorial (4° N - 4° S) sea surface temperature (SST) and 10 m zonal wind fields over the 1950 - 1990 interval. They found the variation consistent with a "devils staircase" interaction between the annual cycle and ENSO. Liu and Duan (2018) use principal oscillation pattern analysis over the 1979 - 2013 era to also identify a QQO (48 months) in global SST anomaly which is dominant in the equatorial Pacific Ocean region. Liu and Xue (2010) investigated the relationship between ENSO and the Antarctic Oscillation (AAO) index with empirical orthogonal function analysis in sea level pressure anomalies from 1951 - 2002. They concluded that ENSO plays a key role in the phase transition of AAO at the quasi-quadrennial timescale. Pisoft et al. (2011) point out that the quasi-quadrennial oscillations that have been reported are almost always associated with the ENSO phenomenon and



variations in sea surface temperatures or the wind field over equatorial areas. Pisoft et al.
(2011) applied a 2-dimensional wavelet transform technique to changes in 500 hPa
temperature fields in two 50-year reanalysis datasets (ERA-40 and NCEP-NCAR) and
established the presence of a distinct QQO and a quasi-decadal oscillation in addition to
the annual and semi-annual cycles. Their analysis showed that the QQO is present in at
least 15 of 50 years in both reanalysis datasets not only in the equatorial zone (30° S to 30°
N), but also over a significant area of the globe including at high latitudes. Both reanalysis
datasets showed relatively high QQO wavelet power north of the Bellinghausen and
Amundsen Seas, over the Bearing Sea and over North America (north of ~45° N). A region
of relatively high wavelet power was detected north of the Mawson and Davis seas near
Antarctica in ERA-40 only.

There are relatively few reports of observations of multi-year variability in the

mesosphere and higher altitudes, and as far as we are aware, the existence of a QQO in the
high latitude mesopause region as discussed here has not previously been reported.
Offermann et al. (2015) reported multi-annual temperature oscillations in Central Europe
detected in SABER data during the period 2002 - 2012, which were reproduced in
simulations by the Hamburg Model of the Neutral and Ionized Atmosphere (HAMMONIA
chemistry climate model (Schmidt et al., 2006) and the Community Earth System Model -
Whole Atmosphere Community Climate Model (CESM-WACCM; Marsh et al., 2013)
models. Periods of 2.4 - 2.2 years, 3.4 and 5.5 years were present in the SABER data over
a range of altitudes from 18 to 110 km. Perminov et al. (2018) reported statistically
significant periods of OH* temperature variations at 3 years and 4.1 years (with amplitudes
of $1.3 \pm 0.2$ K and $0.6 \pm 0.2$ K respectively) from a Lomb-Scargle analysis of 17-years
(2000 - 2016) at Zvenigorod, Russia. Reid et al. (2014) detected significant periodicities
at 4.1 years in $O(^1S)$ emission intensity and ~ 3-year in the OH emission intensities by



performing Lomb-Scargle analysis of a 15 year series of observations at Adelaide,
Australia. However, Perminov et al. (2018) and Reid et al. (2014) do not offer causes of
these periodicities.

The outline of this paper is as follows. In section 2 we review the Davis OH

rotational temperature measurements (described in Part 1 of this study) and the derived
residual temperatures which contain the QQO feature. In section 3 we explore correlation
and composite analyses of the QQO signal using satellite and meteorological reanalysis
datasets. Discussion of the results, summary and conclusions drawn are given in Sections
4 and 5. Additional figures are presented in the supplementary material

As for Part 1 we use the following terminology for the analysed temperature series.

From the measured temperatures and their nightly, monthly, seasonal or winter means,
*temperature anomalies* are produced by subtracting the climatological mean or monthly
mean (we fit a solar cycle and linear long-term trend to the anomalies), *residual*
*temperatures* additionally have the solar cycle component subtracted and *detrended*
*temperatures* have both the solar cycle component and the long term linear trend subtracted.

## 2. Data Sets

2.1 OH(6-2) rotational temperatures

Scanning spectrometer observations of the OH airglow (6-2) band have been made

at Davis station, Antarctica (68.6° S, 78.0° E) for each winter season over the last 24 years
(1995-2018) to provide a time-series of rotational temperatures (a layer weighted proxy of
atmospheric temperatures near 87 km altitude). The solar cycle and long term linear trend
in this temperature series are examined in Part 1 of this study (French et al., 2019).  Fitting
a solar cycle (using 10.7 cm solar flux) and linear long term trend model to the winter mean



temperature anomalies (nightly mean temperatures averaged over day-of-year 106 to 259
with mean climatology subtracted) yields a solar cycle response coefficient $S$ of $4.30 \pm 1.02$
K/100 sfu (95% confidence limits 2.2 K/100 sfu $< S <$ 6.4 K/100 sfu) and a long term linear
trend $L$ of -1.20 ± 0.51 K/decade (95% confidence limits -0.14 K/decade $< L <$ -2.26
K/decade). However, only 58% ($R^2$) of the year-to-year variability is described by this
model (see Fig. 3 of Part 1). Residual temperatures (solar component removed) are shown
in Fig. 1a and the detrended temperatures (solar component and long-term linear fit
removed) in Fig 1b. The QQO signal is apparent, with a peak-peak amplitude of 3-4 K. A
sinusoid fit to the residual temperatures has a peak-to-peak amplitude of 3.0 K and period
of 4.2 years. A wavelet analysis of the residual time-series shown in Fig. 1c reveals an
oscillation period increasing from ~3.5 years in 2000 to 4.5 years in 2013.

We have attempted to examine the seasonal variability of the QQO signal by

dividing averages into intervals FMA, MJJ, ASO (also plotted in Fig 1b). While these
shorter term averages obviously suffer from greater uncertainty, there is a suggestion that
the QQO is strongest over the winter months MJJ, mid-range in ASO and less apparent in
the FMA interval.

2.2 Aura/MLS temperature profiles

Long-term temperature data for the mesopause region are available from two

satellite instruments; the Microwave Limb Sounder on the Aura satellite (Aura/MLS) and
the Sounding of the Atmosphere using Broadband Emission Radiometry (SABER)
instrument on the Thermosphere Ionosphere Mesosphere Energetics Dynamics (TIMED)
satellite. Hydroxyl layer equivalent temperature measurements from these instruments are
overlaid on Fig 1a.



Aura/MLS temperatures are derived from observations of thermal microwave
emissions near the oxygen spectral lines of $O_2$ (118 GHz) and $O^{18}O$ (234 GHz). The
instrument scans the Earth's limb every 24.7 s and the retrieval algorithm (for version v4.2
level 2 used here) produces useful temperature profiles on a fixed vertical pressure grid
from 316 hPa (~8 km) to 0.001 hPa (~97 km) over the latitude range 82° S - 82° N with
about 14 orbits per day.   The along-track resolution is typically 165 km through the
stratosphere to 220 km at the top of the mesosphere. The vertical resolution is defined by
the full width at half maximum of the averaging kernels and varies from 5.3 km at 316 hPa
to 9 km at 0.1 hPa and up to 15 km at 0.001 hPa (Schwartz et al., 2008). We also use
profiles of carbon monoxide (CO) mixing ratio, which are scientifically useful between
215 hPa to 0.0046 hPa and have similar vertical and horizontal resolution to the temperature
measurements.
For comparison with the Davis OH temperatures we retrieved Aura/MLS profiles
acquired within 500 km of Davis station (about 60 coincident samples per month) between
2005 (Aura launched in July 2004) and 2018, applied selection criteria according to the
quality control recommendations described in Livesey et al. (2018) and averaged over the
winter months April to September (AMJJAS; similar to the averaging period for the Davis
winter mean) at the native Aura/MLS retrieval pressure level of 0.00464 hPa (0.46 Pa).
The 0.00464 hPa pressure level statistically provides the best fit, in absolute temperature
terms, and in the correlation of variability, to the OH(6-2) temperatures we derive using
Langhoff et al. (1986) transition probabilities. The Aura/MLS AMJJAS mean temperature
residuals are overlaid in Fig. 1a (green line from 2005); the solar cycle component is
removed in the same way as for the OH data. For comparison, regression values for
Aura/MLS (2005 – 2018) are $3.4 \pm 2.3$ K/100 sfu for the solar term and $-1.3 \pm 1.2$ K/decade
for the long term trend (but neither term is significant at the 95% level; $R^2 = 0.2$). If the





solar response coefficient derived for the 24 years of OH measurements is used to compute
residuals, the Aura/MLS long term linear trend becomes -1.4 ± 1.1 K/decade ($R^2 = 0.12$).
The Aura/MLS measurements show very good agreement with the OH measurements, both
in terms of the long term linear fit to the residuals, and in the magnitude and pattern of the
QQO feature over its last 3 cycles.

2.3 TIMED/SABER profiles
The SABER instrument measures Earth limb emission profiles over the 1.27 – 17
µm spectral range from the TIMED satellite, which was launched in December 2001 into
a circular orbit at 625 km altitude and 74° inclination to the equator (Russell III et al., 1999).
The satellite undergoes a yaw cycle every 60 days, alternating coverage of latitude bands
54° S to 82° N and  82° S to 54° N, and precessing slowly to complete 24 h local time over
the yaw interval. Temperature is retrieved over an altitude range of 10 – 105 km, with a
vertical resolution of about 2 km, and along track resolution of 400 km from 15 µm and
4.3 µm carbon dioxide ($CO_2$) emissions (Mertens et al., 2003). Generally, errors in the
retrieved temperatures in the mesopause region are estimated to be in the range ±1.5 – 5 K
(García-Comas et al., 2008).
SABER also measures a volume emission rate (VER) from a radiometer sensitive
over the 1.56 – 1.72 µm spectral range (OH-B channel) which includes mostly the OH(4-
2) and OH(5-3) bands. SABER v2.0 Level 2B data are used in this study. We use a
Gaussian fit to the VER to derive weighted average OH layer equivalent temperatures
(T_VER; as for French and Mulligan, 2010). While the VER layer weighting function is
not explicitly a vibrational level 6 profile but a combination of the 4 and 5 vibrational levels
from the OH-B channel, the difference from the v'=6 profile in terms of peak altitude is



not expected to be greater than 1 km, compared to the ~8 km full- width at half maximum
(FWHM) of the layer (McDade, 1991; von Savigny et al., 2012).

As for Aura/MLS we average all SABER profiles within 500 km radius of Davis

station that fall within the winter averaging window. Due to the satellite yaw cycle this
limits SABER observations to two intervals day-of-year 75 – 140 and 196 – 262 and the
days prior to 106 and after 259 are rejected as outside the OH winter interval. Essentially,
SABER samples the same as the OH-observations except days 141 - 195 (21 May to 14
July) are excluded.  As for OH temperatures and Aura/MLS a fit of solar cycle (F10.7) and
linear trend terms is made to remove the solar cycle component. Regression values for
SABER for 2002 - 2018 are 3.4 ± 1.8 K/100 sfu for the solar term and -0.77 ± 1.05
K/decade for the long term trend. Neither term is statistically significant at the 95% level
($R^2 = 0.22$).  The solar term is significant at the 90% level.  Both the OH-peak altitude and
the OH T_VER show slightly negative trends over the period 2002 - 2018 (-0.02 ± 0.02
km/year $R^2 = 0.09$ and -0.13 ± 0.11 K/year $R^2 = 0.09$ respectively) but they are not
statistically significant. There is a slight anti-correlation between OH-peak altitude and
T_VER (-2.2 ± 1.7 K/km, $R^2 = 0.1$), but once again, the value is not statistically significant.
Plots of the OH peak altitude and OH T_VER time series and the anti-correlation
relationship are provided in Fig. S1 of the supplementary material.

The derived SABER residual mean temperatures are also plotted in Fig. 1a (pink

dotted line) for years 2002 - 2018. Given that the yaw cycle excludes days 141-195 from
the winter averaging in these data, in general the SABER residual temperatures also
reproduce the QQO variation, except 2011 appears to be anomalously warm (by ~3 K).
The OH peak altitude derived from the SABER OH_VER profile also shows an
anomalously low layer altitude for 2011 (lowest winter mean altitude in the 2002 - 2018
year record)




2.4 ECMWF/ERA5

As discussed in the introduction, reported oscillations on a quasi-quadrennial scale

have almost always been associated with ENSO and its interactions in near-surface
equatorial pressure, wind and sea surface temperature fields.  To investigate the possible
connection to the Antarctic mesopause QQO observation we perform correlation and
composite analyses using the European Centre for Medium Range Weather Forecasting
(ECMWF) ERA5 reanalysis products (Copernicus Climate Change Service, 2017;
https://apps.ecmwf.int/data-catalogues/era5/?class=ea). These include global monthly
average geopotential height and wind components provided on 37 pressure levels (surface
to 1 hPa) at 0.25° x 0.25° grid point resolution.

2.5 ERSST and OISST

Sea surface temperatures (SST) used in this study are from the National Oceanic

and Atmospheric Administration (NOAA) Extended Reconstructed Sea Surface
Temperature (ERSST v5;  https://www.ncdc.noaa.gov/data-access/marineocean-data/
extended-reconstructed-sea-surface-temperature-ersst-v5) monthly dataset derived from
the International Comprehensive Ocean–Atmosphere Dataset (ICOADS). These are
available globally extending from January 1854 to the present at 2° x 2° grid resolution
with spatial completeness enhanced using statistical methods (Huang et al., 2017).

For sea ice cover, we also use the NOAA Optimum Interpolation Sea Surface

Temperature (OISST) V2 product, available as monthly means on a 1° global grid  using
in situ and satellite SSTs plus SSTs simulated by sea-ice cover.(Reynolds et al., 2002;
https://www.esrl.noaa.gov/psd/data/ gridded/data.noaa.oisst.v2.html )





## 3. Features of the QQO


The spatial extent of the QQO signal observed at Davis is explored with the
Aura/MLS dataset for the 2005 - 2018 interval of concurrent observations. These data are
averaged into 5° x 10° (latitude x longitude) grid cells. In Fig. 2a) we correlate the
Aura/MLS AMJJAS temperature residual time series for the grid cell over Davis (14 years;
time series plotted in left hand panel) with each grid cell of the Aura/MLS AMJJAS
temperature residual at 0.0046 hPa. The 3 map panels show correlation ($R$) coefficient in
equi-rectangular, Southern Hemisphere (SH) and Northern Hemisphere (NH) projections.
The correlation colour scale is common to all maps and hashed areas show significance at
the 90% level. The Davis QQO signal shows a significant positive correlation with a large
part of the east Antarctic and southern Indian Ocean sectors and significant anti-correlation
in the Southern Ocean near New Zealand. In the NH summer, there is a general region of
negative correlation at mid- to high-latitudes, indicating that the QQO has opposite phases
in the two hemispheres. We return to examine phase response between the hemispheres in
Section 4.
Extending this analysis, Fig. 2b shows the correlation between the mean
temperature of the polar cap (65° S - 85° S), which has a variation similar to that shown in
Fig. 2a for Davis (left hand panel), with Aura/MLS temperature for each grid box on
different pressure levels. It is apparent that the QQO signal observed at Davis extends over
the majority of the polar cap, and through most of the mesosphere down to at least the 0.1
hPa level with similar amplitude (3 - 4 K peak-to-peak) and phase. Significant anti-
correlation of the QQO signal then occurs in the upper stratosphere (pressure range 1 - 10
hPa) in the polar cap and Southern Ocean, while a significant positive correlation occurs
in the region of the subtropical jets at 10 hPa.





258   We further examine the Davis QQO signal using correlation and composite analysis

259 with the ECMWF/ERA5 reanalysis data and NOAA/ERSST v5 data described above.

260 Figure 3 shows the composites of the ERA5 geopotential height anomaly (with respect to

261 the 1995-2018 climatology) averaged over AMJJAS for the 33$^{rd}$ percentile ('cold' years),

262 the 67$^{th}$ percentile ('warm' years) and the remaining 'mid' years (between the 33$^{rd}$ and 67$^{th}$

263 percentiles) of the *detrended* Davis hydroxyl temperature winter average QQO signal at

264 pressure levels of 750, 200, 50, 10 and 1 hPa. The first two pressure levels are globally

265 generally within the troposphere, and the other levels are generally within the stratosphere.

266 The 'cold' years (threshold -0.99 K) and their detrended temperature values (in parenthesis)

267 are: 1997 (-1.166), 2001 (-3.039), 2005 (-1.023), 2008 (-1.188), 2009 (-1.738), 2010 (-

268 1.998), 2014 (-1.381), 2018 (-2.451). The 'warm' years (threshold 1.24 K) and their

269 detrended temperature values (in parenthesis) are: 1996 (2.325), 1999 (1.399), 2002 (1.523),

270 2007 (2.210), 2011 (1.796), 2012 (1.241), 2015 (1.688), 2016 (2.287). The cold (warm)

271 years are shown by the blue (red) dots in Fig. 1b. Composite maps for meridional and zonal

272 wind anomalies at these pressure levels are provided for comparison in Figs. 4 and S2,

273 respectively. The hatching used in these figures to indicate significance is set at the 90%

274 level based on a two-tailed Student's T-test assuming normally distributed statistics.

275   Examining Fig. 3, it is seen that cold years are associated with a small significant

276 region of negative geopotential height anomaly to the north-east of Davis at 750 hPa, which

277 expands and appears to shift equatorward and westward at higher altitudes (lower

278 pressures). A similarly placed, though more extensive region of significant positive

279 geopotential height anomaly is seen in the composites for the warm years, and generally

280 the mid- to high-latitude regions of significant anomalies appear to have opposite signs

281 between the composites for the cold and warm years. Note that in the northern high-

282 latitudes there are regions of negative geopotential height anomaly in the cold year


composites, and similarly placed positive anomalies for the warm year composites,
although these are not significant in all cases. The intermediate composites provide some
contrasts with the cold and warm year composites. For the intermediate years at and below
the 50 hPa level, there are negative anomalies in the southern polar cap and mid-latitudes
that are similarly located to positive anomalies for the warm years. However at 10 hPa and
1 hPa, the main negative feature at southern high latitudes in the intermediate composites
is in the southern Pacific Ocean towards the Antarctic coasts.
In Fig. 4, the 50, 10 and 1 hPa levels show large-scale patterns in mid- and high-
latitudes of the SH. For the cold year composite, there is a region of negative (poleward)
anomalous flow south of Australia towards Antarctica at 50 hPa that extends further
poleward and westward towards and over Davis in the upper levels. At 1 hPa, the pattern
generally has a zonal wave-1 structure, which is also seen in the cold year geopotential
height anomaly composite at this level (top left panel of Fig. 3). The warm year meridional
wind composite (right panels of Fig. 4) appears to show a pattern that has a wave-2 structure
in the upper levels, with a general orientation north west to south east at 1 hPa. In the region
near Davis, the meridional wind anomaly is equatorward in the upper levels for the warm
years. The intermediate years provide a contrast between the cold and warm years in the
meridional wind, with regions that show opposite sign in some of the regions common with
either the cold or warm year composites. For example, at 1 hPa near Australia, the
meridional wind anomaly is positive (equatorward) for the intermediate years and negative
(poleward) for the warm years, whereas near the Antarctic Peninsula, the wind anomaly is
positive in the cold years and negative in the intermediate years. The zonal wind composites
(provided in supplementary Fig. S2) also show contrasting patterns between the cold,
intermediate and warm year averages, particularly in the upper levels. Here regions of





significant anomalies tend to extend into the tropics and NH. At 10 hPa the SH patterns
tend to show a wave-3 structure in the cold and warm years.

Overall there are geographical regions showing some clear anti-phase relationships

between the cold and warm years in the ERA5 composites (i.e. statistically significant and
of opposite sign), but the intermediate years also show significant patterns suggesting that
there may not be one clear driver for any association with the mesopause region
temperatures. Some intermediate years are close to the cold or warm threshold, but varying
the threshold did not significantly alter the patterns in the composites. We also produced
composite maps using ERA5 temperatures (not shown). While able to span more years than
that possible with Aura/MLS temperatures, the ERA5 composites for the 1 hPa and 10 hPa
levels were qualitatively consistent with the correlation maps shown in the lower two rows
of panels in Fig. 2b.

Figure 5 presents correlation maps of both the Davis OH residual winter average

QQO signal (24 years; a) and Aura/MLS 0.0046hPa polar cap [AMJJAS] residual QQO
signal (14 years; b) against ERSST v5 anomalies (evaluated with respect to the 1995-2018
climatology). The strongest and most consistent patterns of anti-correlation (QQO warmest
for below average SST) for the two epochs occur at mid-latitudes in the south-western
Pacific Ocean (to the south of Australia and New Zealand), in the south-western Atlantic
Ocean (near the east coast of South America), and in the west-central Indian Ocean (to the
west of Madagascar). Significant positive correlation is also seen at mid-latitudes south of
Africa, and for the longer-term Davis data set, in the south-eastern Pacific Ocean. The
correlation maps generally show a dipole-like pattern in the Indian Ocean (although the
positive correlation in the east-central Indian Ocean is not significant), and weak or no
correlation in the central Pacific Ocean where ENSO SST anomalies tend to be located.
Comparing with the 500 hPa air temperature analysis of Pisoft et al. (2011), their Fig. 3





shows regions of high wavelet power in the QQO timescale at mid- and high southern
latitudes for the ERA-40 reanalysis that bear some similarity to the location of regions of
high correlation in Fig. 5a.





## 4. Discussion

### 4.1 Antarctic sea ice

On the basis of the SST patterns apparent in Fig. 5, we examined the possibility of a QQO signal in Antarctic sea ice concentration using the NOAA Optimum Interpolation SST (OI-SST) dataset version 2 (Reynolds et al., 2002). As can be seen in Fig. 6a, there are regions of significant negative correlation between the Davis OH residual temperature time series and OI-SST sea ice concentration towards the Antarctic coast between $30°$ E and $60°$ E (south east of Africa), and also centered on $120°$ W (in the Amundsen Sea). These regions tend to lie to the south of regions where the SST is positively correlated with the Davis OH temperature residuals (Fig. 6b), consistent with warm (cold) SSTs having reducing (increasing) sea ice concentration. A link between sea ice concentration and meridional and zonal wind could be expected if persistent near-surface circulation anomalies are related to the mesospheric QQO. For example, a persistent northward (southward) flow on one side of a circulation anomaly could increase (decrease) sea ice due to the associated flow of relatively cold (warm) air from higher (lower) latitudes and expansion (compaction) of the ice edge (e.g. Fig. 3 of Turner et al., 2016). Both the zonal and meridional near-surface (10 m) wind components (Fig. 6c and 6d) show regions of negative correlation with the Davis OH temperature QQO at the Antarctic coast near 30 - 60 ° E where the pattern in Fig. 6a is significant, but these correlations are generally weak and of relatively small area. There is also a weak and not significant positive correlation around much of the equatorward edge of the sea ice zone.

Further equatorward from the Antarctic coast, Fig. 6c and 6d show correlation patterns (marked 'A', 'B' and 'C' on Fig. 6c) that are suggestive of cyclonic (anticyclonic) circulation under warm (cold) QQO phases. These features appear to be consistent with the extent of negative (cyclonic) and positive (anti-cyclonic) geopotential height anomalies at



200 hPa and 750 hPa in the warm and cold composites of Fig. 3, respectively (also marked
'A','B' and 'C' on Fig 3. bottom right panel). Intriguingly, the features appear close to the
'gatekeeper' circulation features in the southern Indian Ocean (SIO), south-west Pacific
Ocean (SWP) and south-west Atlantic Ocean (SWA), respectively, identified by Turney et
al. (2015) as having a strong influence on Antarctic surface temperatures (see their Fig. 4).
This could hint as to the origin of the QQO forcing as residing with a tropical interaction
with  the mid- and high latitude SH circulation, particularly in the wave-3 near-surface
features of the southern high latitudes (Raphael, 2004) which potentially also influences
surface conditions including sea ice. Furthermore, we note that Turney et al. (2015) in their
Fig. 9 show periodicities in South Pole temperatures in the 4 – 6 year period in recent
decades that appear to be associated with the variations in pressure in the SIO and SWP
regions.

Figure 4c of Parkinson (2019) shows the annual time series of Antarctic sea ice

extent in the Indian Ocean sector (20° E – 90° E) which spans the general region of negative
correlation at eastern longitudes in Fig. 6a. While there is evidence for an anti-correlation
of this sea ice time series with the Davis OH temperature residuals (Fig. S3; $R^2 = 0.26$, p =
0.001), it is not consistent for all years (e.g. 1999). Parkinson (2019) also provides a sea
ice time series for the Ross sea region (160° E – 130° W), which covers part of the region
of significant negative correlation in Fig. 6a, but also a more extensive region to the east.
There is a weak but not significant negative correlation between this time series and the
Davis OH temperature residual ($R$ = -0.09).

Overall, further investigation of sea ice variability in connection with a QQO

signature is suggested, particularly as the annual time series of Antarctic sea ice extent
presented in Fig. 1c of Parkinson (2019) appears to show 4 - 6 year variability, at least





since the early 1990s, which generally appears anti-correlated with the QQO temperature
signal.

4.2 QBO and ENSO relationships to the QQO

Residual variability in the Davis OH data set has been previously investigated by

French and Klekociuk (2011) using indices for planetary wave activity (derived by zonal
Fourier decomposition of the 10 hPa geopotential height at 67.5° S from the NCEP-DOE
reanalysis, polar vortex intensity (PVI based on the zonal wind anomaly at 10 hPa), 30 hPa
standardized QBO and the Southern Annular Mode (SAM; calculated as the difference
between the normalized monthly mean sea level pressure at 45° S and 65° S). No
statistically significant correlations were found with the PVI, QBO or SAM indices over
the entire data set (then extending 1995 - 2010); however, there was clear evidence of
planetary wave modes identified in the 10 hPa NCEP reanalysis data penetrating to OH
layer heights at different times in the series. Over the shorter time series, the QQO was not
readily apparent in that study.

In a study of the SH summer mesosphere responses to ENSO, Li et al. (2016)

suggest that constructive interference of ENSO and QBO could lead to stronger
stratospheric westward zonal wind anomalies at SH high-latitudes in November and
December thereby causing early breakdown of the SH stratospheric polar vortex during
warm ENSO events in the westward phase of the QBO.  This would in turn lead to greater
SH mesospheric eastward gravity wave (GW) forcing and much colder polar temperatures.
The opposite effect would occur during cold ENSO events in the eastward QBO phase
leading to warmer mesospheric polar temperatures.  We have re-examined the QBO and
ENSO indexes as likely candidates for a possible source of the observed QQO. However,
comparing both 30 hPa and 10 hPa Singapore QBO data (https://www.geo.fu-





berlin.de/en/met/ag/strat/produkte/qbo/) and the Multivariate ENSO index (MEIv2)
(https://www.esrl.noaa.gov/psd/enso/mei/) yields no significant correlation to the QQO
variation. Time series plots are available in the Figs. S4 and S5 of the supplementary
material.

The clear presence of patterns in the ERA5 composite data in Fig. 3 (wave-1

structure at 1 hPa, particularly in the cold years), and Fig. 4 (wave-2 structure at 1 hPa in
the warm years) suggests that non-migrating tides or stationary planetary waves may have
some part to play in the formation of the QQO. Baldwin et al. (2019) reported strong inter-
annual variability in the amplitude of the diurnal migrating tide (DW1) observed in SABER
temperature data, which appears to be related to the stratospheric QBO. Liu (2016) notes
that the modulation of tides by the QBO and ENSO can have an impact at inter-annual
timescales in a review of the influence of low atmosphere forcing on variability of the space
environment. The absence of a direct correlation between the QQO and the QBO (or
ENSO), together with the presence of distinctly different wave patterns in the ERA5
geopotential and meridional wind anomalies during warm and cold years of the QQO,
provide a tantalizing picture of the complexity of the mechanisms that influence the upper
atmosphere.

4.3 Relationship with Mesospheric Zonal and Meridional Winds

To further explore the origin of the QQO temperature variation, we examined the

AMJJAS mean meridional and zonal winds measured by the medium frequency spaced
antenna (MFSA) radar, co-located at Davis (Murphy et al., 2012). Correlations between
the Davis OH and SABER residual temperatures (compared over the common satellite era
2002-2018) and the MFSA meridional wind at 86 km both yield $R^2$ values of 0.51 as shown
in Fig. 7. The Aura/MLS correlation $R^2$ is 0.54 over the shorter time span (2005-2018). The



correlation between mesopause region temperature and the meridional wind is such that
higher (lower) temperatures correspond to poleward (equatorward) flow over the site.
There is no significant correlation with the zonal wind.

Dyrland et al. (2010) and Espy et al. (2003) have reported a similar relation

between temperature and the background wind in the mesosphere, which they explain in
terms of adiabatic processes whereby poleward circulation leads to convergence and
downwelling and therefore adiabatic heating, while equatorward circulation is
symptomatic of upwelling and cooling.  The correlation suggests that at least part of the
temperature variation at Davis after removal of the seasonal cycle, the solar cycle response,
and the long-term linear trend is due to the adiabatic action of the residual meridional
circulation.  This hypothesis is supported by the Aura/MLS polar cap correlation plots
which show the highest correlation in the region of the SH polar cap, but only down to an
altitude of ~64 km (0.1 hPa).

4.4 CO as a tracer of vertical transport

Additional evidence supporting the association between temperature and large-

scale adiabatic processes over the polar cap was obtained by examining the concentration
of CO using Aura/MLS measurements.  The primary source of CO in the upper stratosphere
and mesosphere is photolysis of $CO_2$, while production via oxidation of methane occurs
throughout the middle atmosphere (Brasseur and Solomon, 2005; Lee et al., 2018).  The
long lifetime (> 1 month) of CO makes it a useful tracer for vertical and meridional
transport, particularly during the polar winter when there is a lack of photolysis over the
polar cap.  Figure 8 shows a general positive correlation between the time series of the SH
polar cap winter residual temperature at 0.0046 hPa and CO mixing ratio at levels between
0.0046 hPa and 0.1 hPa using Aura/MLS data. Here we have used the same gridding and



averaging as for Fig. 2, and obtain the residual by subtracting the seasonal cycle and a fitted
solar cycle response for each grid box before forming averages over the polar cap. The
linear correlation coefficient at 0.0046 hPa between the temperature and CO time series is
0.11, which increases to 0.43 ($R^2 = 0.13$) on removal of the linear trends from both time
series (-0.74 K/decade in temperature and +0.65 ppmv/decade or ~+0.4% per decade).
Similar values of $R^2$ are observed down to the 10 hPa level on removal of linear trends. The
positive correlation is consistent with CO being transported into (out of) the polar cap by
convergence (divergence) of air masses which cause adiabatic warming (cooling) in the
process.  As there is a strong positive vertical gradient in CO in the upper mesosphere (Lee
et al., 2018), we suggest that the largest contribution to changes in CO in Fig. 8 is from
vertical transport rather than from horizontal transport. Below the 0.1 hPa pressure level,
the correlations diminish.

Figure 9 shows the spatial correlation between the polar cap QQO temperature

signal at 0.0046 hPa and the CO residual mixing ratio at four pressure levels. In Fig. 9a,
temperature and CO are significantly positively correlated over most of Antarctica, which
is consistent with Fig. 8 and our hypothesis that the QQO temperature variation is an
adiabatic response (i.e. increased (decreased) temperature is associated with increased
(decreased) CO concentration due to descent (ascent)). This general positive correlation
over Antarctica is also seen for CO at 0.1 hPa (Fig. 9b) and is apparent though less clear
for CO at 1 hPa (Fig. 9c). There are also regions of significant positive correlation to the
south east of Madagascar and near the southern tip of South America that are consistent
with the regions of significant positive correlation in Fig. 2b. Note however that the region
of significant negative correlation in Fig. 2b south of New Zealand also shows negative
correlation in Fig. 9a, albeit mostly not significant. This suggests that temperature and CO
are in opposing phases in this region, unlike the in-phase response over Antarctica. The





implication here is that the response in the sub-New Zealand region does not tally with an
adiabatic response. In the NH, there are generally no large scale significant patterns of
correlation. However it can be seen, particularly in Fig. 9a, that there are regions of
significant correlation between the SH polar temperature and CO over northern mid- to
high-latitudes, which have negative correlation in Fig. 2b. This suggests that the QQO in
the NH summer is in the opposite phase to the QQO in the SH winter, and that the forcing
of temperature is consistent with an adiabatic response.

4.5 Interhemispheric coupling

Returning to the phase response of the QQO signal between the hemispheres, we

performed a similar analysis to that shown in Fig. 2 but using average temperatures across
the Arctic polar cap (65° N to 85° N) in winter months October to March (ONDJFM) and
summer months (AMJJAS). We compare the time series in Fig. 10, together with SH polar
cap summer and winter time series. First we see that in winter, the NH response (red line
with linear fit) has a somewhat smaller amplitude than in the SH (green line with linear fit)
and a less clear QQO variation. For the SH summer (dark green line), the response is
generally in-phase with the SH winter, except in years 2005-2008. In addition, the
amplitude of the SH summer response is larger than for the SH winter. For the NH summer
(orange line), the response is in an approximately opposite phase to the SH winter (except
for years 2005-2006), but with similar amplitude to the SH summer. Our QQO signal for
the NH summer polar cap shown in Fig. 10 is consistent with temperatures for 2002 - 2010
NH summers shown in Fig. 5 of Russell et al. (2014) poleward of 60° N using
TIMED/SABER and Aura/MLS data.

4.6 Comparison with CESM-WACCM





Following on from Offermann et al. (2015), we examined simulations produced
from a version of CESM-WACCM for phase 1 of Chemistry-Climate Model Initiative
(CCMI-1; (Morgenstern et al., 2017)), specifically the CESM1 WACCM model which
includes both interactive atmospheric chemistry and interactive ocean physics to provide a
self-consistent simulation of climate. Our interest here is to see if the model physics
produces a QQO response in the mesosphere. We obtained 3 ensemble members from the
REF-C2 simulation of the model spanning 1955-2100. The REF-C2 simulation follows
particular scenarios for interactive chemistry involving ozone depleting substances and
radiative forcing (the WMO A1 and RCP 6.0 scenarios, respectively; Morgenstern et al.,
2017). SH polar cap (65° S - 85° S) temperatures were averaged over AMJJAS for pressure
levels of 0.01 Pa (~105 km altitude), 0.5 Pa (~85 km) and 0.15 hPa (~60 km), and Morlet
wavelet analysis (Torrence and Compo, 1998) was applied to the individual ensemble
members. While a solar-cycle (10 - 11 year period) signal was detected with better than
95% confidence at each pressure level for all ensemble members, no periodicity in the 3 -
6 year range exceeded the 95% confidence limit for any member.

4.7 Gravity wave interaction
We now consider why our SH winter QQO signal appears generally restricted to
the mesosphere. It is well known that the Antarctic Peninsula is a hot spot for gravity wave
activity at the edge of the southern polar cap (Hoffmann et al., 2013) and this region is
consistently active during austral autumn, winter and spring. Many GWs are able to
penetrate all the way up to the mesosphere before their amplitudes become so large that
they break, and deposit their energy, thereby introducing perturbations in winds and
temperatures. Correlation coefficients below the 0.1 hPa level (~65 km) in our Aura/MLS
analysis are potentially low because few GWs break below this altitude. The strong





eastward stratospheric winds of the polar night jet filter out many eastward propagating
GWs creating a westward drag on winds in the mesosphere, which when combined with
the Coriolis force generates a weak poleward flow. This flow is modulated by interannual
variations in upward propagation of GWs and agrees with the view of (Solomon et al.,
2018) who attribute the significant inter-annual variability of mesopause temperatures to
the dominance of dynamical processes in their control. Further support for this view can
be found in (Sato et al., 2012) who employed a high-resolution middle atmosphere general
circulation model (GCM) to examine gravity wave propagation in the middle to high
latitudes of the SH without the need for gravity wave parameterization. Gravity wave
energy is generally weak in summer but in winter, gravity waves have large amplitudes
and are distributed around the polar vortex in the upper stratosphere and mesosphere. The
wave energy is not zonally uniformly distributed but is concentrated on the leeward side of
the Southern Andes and Antarctic Peninsula. Energy propagation extends several thousand
kilometres eastwards which explains the gravity wave distribution around the polar vortex
in winter.
Examining the Aura/MLS polar cap correlation plots in Fig. 2(a) and 2(b) in detail,
we see that maps of GW potential energy (PE) at 10 hPa calculated for the winter months
by Sato et al. (2012; their Fig. 2) is well reproduced at 0.1 hPa, and that the region of
highest correlation becomes more concentrated as GWs are filtered out with increasing
altitude. It is also likely that GWs are strongly focussed into the polar night wind jet
(Wright et al., 2017). Wright et al. (2016) reported strong correlations between GW
potential energy and vertical wavelength with stratospheric winds, but not local surface
winds from a multi-instruments gravity-wave investigation over Tierra del Fuego and the
Drake Passage.



4.8 Mechanisms for a 4 year cycle

The question remains as to why does this modulation have a quasi-four year cycle? Zhang et al. (2017) detected both three- and four-year oscillations in zonal mean SABER temperatures at 85 km altitude in the period 2002-2015 using Lomb-Scargle analysis, in which the much stronger annual, semi-annual, quasi-biennial, and 11-year periods were also present. The latitude range studied was limited to 50° S to 50° N because of the satellite yaw cycle; the four-year oscillation was found to have a stronger peak in the SH. Although the origin of the four-year oscillation is not discussed in Zhang et al. (2017), it is suggested that the three-year oscillation is a sub-peak of the QBO, and is due to modulation of the QBO possibly by the semiannual oscillation. Their analyzed SABER temperatures also show evidence of the four-year oscillation at 25 km altitude, but not at 45 km. We note that a QQO variability observed in Jupiter's equatorial winds has been inferred to result from forcing by gravity waves produced by deep convection (Cosentino et al., 2017).

Liu et al. (2017) examined variations in global gravity waves from 14 years of SABER temperatures between 2002 and 2015. Unfortunately, their study was limited to the latitude band 50° S to 50° N because of the TIMED 60-day yaw cycle. They applied multivariate linear regression to calculate trends of global GW potential energy and the responses of GW PE to solar activity, to the QBO and to ENSO. They found a positive trend in GW PE with a maximum of 12-15% per decade at 40° S - 50° S below 60 km altitude. This was interpreted as a possible indication of eddy diffusion increase in some locations, and at 50° S could be due to a strengthening of the polar stratospheric jets. Increasing eddy diffusion was advanced as a possible explanation of increasing $CO_2$ trends with altitude (Emmert et al., 2012). However, Qian et al. (2019) have shown that sampling of SABER data in window lengths less than 60 days can lead to incorrect $CO_2$ values. As a result, increased eddy diffusion is no longer necessary to explain the anomalous $CO_2$



result. The global gravity wave response to solar activity is negative in lower and mid-
latitudes in the mesosphere lower-thermosphere (MLT) region. It is also negative to the
QBO eastward wind phase in the tropics, and is more negative in the NH than in the SH
MLT region. The response of global GWs to the ENSO index is positive in the tropical
stratosphere (Geller et al., 2016).

Yasui et al. (2016) examined the seasonal and inter-annual variations of GWs (50 -

100 km) using an MF radar at Syowa Station (1999-2013). They found that the Antarctic
summer inter-annual modulation could not be explained by the proposed mechanism of
SSWs in the Arctic via inter-hemisphere coupling. Two other proposed mechanisms were
found to be the more likely origin of the modulation: these were: (a) modulation of the
vertical filtering of GWs in association with breakdown of the polar vortex in the SH, and
(b) tropical convection and propagation to the Antarctic region. The periods noted in the
Introduction in the study of the mesosphere over Central Europe reported by Offermann et
al. (2015) fit well with the correlation results for Aura/MLS temperatures at different
pressure levels in Fig. 2 of the present study. The amplitude of the oscillations they report
(~1 K) are about half those observed in the QQO at Davis. In addition, they state that these
type of oscillations are found in the GLOTI (Global Land Ocean Temperature Index) and
NAO (North Atlantic Oscillation index) data, which supports the correlation results with
the SSTs observed in this work. Concerning possible origins of these oscillations,
Offermann et al. (2015) suggest harmonics of the 11-year solar cycle at 5.5 years, 3.6 years
and 2.2 years, in addition to synchronisation of adjacent atmospheric layers acting as
independent non-linear oscillators.

## 5. Summary and Conclusions

The variability in temperatures derived from hydroxyl airglow observations at Davis Station are examined after seasonal, solar cycle and long-term linear trend terms are removed (in Part 1 of this work, French et al., 2019). The following observations are made regarding this variability:

- A strong QQO feature (3-4 K peak-to-peak amplitude, 3.5 - 4.5 year period) has been observed in the mesopause region temperatures measured at Davis research station which has been sustained over more than two solar cycles (24 years).

- Previous reports of QQO signals have tended to be associated with the ENSO phenomenon and sea surface temperatures, or the wind field over equatorial regions, but this is the first report of its presence at high latitude mesopause altitudes.

- Observations from both Aura/MLS (from 2005) and TIMED/SABER (from 2002) support the Davis QQO feature in amplitude, period and phase.

- Correlation of the QQO pattern detected at Davis with the Aura/MLS global temperature field at 0.0046 hPa shows that the QQO has a significant positive correlation with a large part of the Antarctic polar cap and southern Indian Ocean sectors and significant anti-correlation in the Southern Ocean below New Zealand. The polar cap average (65° S - 85° S) has a very similar QQO pattern to the Davis site. There is a general region of negative correlation at mid- to high latitudes, in the Northern Hemisphere (NH) summer, indicating that the QQO has opposite phases in the two hemispheres.

- Correlation of the SH polar cap average QQO signal shows that the pattern extends vertically from the mesopause region (~86 km) down to 0.1 hPa (~64



km) and then becomes anti-correlated in the upper stratosphere (1 – 10 hPa).
Again, this pattern is opposite in the NH.
•  Composite analysis with ERA5 geopotential anomaly indicate warm years of the
QQO are associated with higher than average geopotential height anomalies over
the polar cap, the East Antarctic sector of the Southern Ocean (sub-Africa) and
the Amundsen Sea region and lower than average anomalies in the southern
Pacific, Indian and Atlantic regions. Cold years are associated with the opposite
and the effect is greater at higher altitudes (10 and 1 hPa levels). There is the
indication of a connection with persistent near-surface circulation anomalies in
the southern Indian Ocean and south-west Pacific Ocean, and variability in
Antarctic sea ice.
•  Composite analysis with ERA5 data also indicates the presence of distinctly
different wave patterns in the ERA5 geopotential and meridional wind anomalies
during the warm and cold years of the QQO, indicative of a potential role of
planetary waves or atmospheric tides in the QQO.
•  Correlation with the meridional wind anomaly at 86 km measured by the Davis
medium frequency spaced antenna radar shows that about 51% of the mesopause
temperature QQO can be explained by the adiabatic cooling (heating) resulting
from meridional circulation. This result is supported by the anti-correlation
between temperature and Aura/MLS CO measurements on a global scale as
reported by Lee et al. (2018).
•  The modulation of the meridional circulation is most likely a result of the
variation of the gravity wave filtering by the strong stratospheric winds during
the polar night.





Taken together, these points highlight the interconnectedness of the entire
atmosphere-ocean system, and that the QQO may be a manifestation of some type of
normal oscillatory atmospheric mode arising from atmosphere-ocean interactions. Our
efforts to isolate a specific mechanism that would drive a QQO, such as combinations of
ENSO, QBO, PVI, SAM (like those proposed by Li et al. (2016) in the SH summer) have
not found anything definite in the winter data at this time, and further investigations are
warranted.
As we have shown, the QQO signal is also present in the polar summer mesosphere,
and consequently there are implications for multi-year variability in the summer polar
phenomena of noctilucent clouds (or Polar Mesospheric Clouds (PMC)), and Polar
Mesospheric Summer Echoes. It would be expected that the temperature perturbations of
3 – 4 K that accompany the QQO at the mesopause will tend to have the most significant
influence where temperatures hover near the ice-aerosol formation threshold, perturbed by
gravity waves, planetary waves and tides. Indeed, the QQO signal may explain part of the
variability in the position of the low-latitude boundary and modelled occurrence of
noctilucent clouds in the NH reported by Russell et al. (2014), and the albedo of PMC for
the SH and NH reported by Liu et al. (2016). The implications of the QQO for long-term
trends in these mesospheric phenomena deserves further study.

## 675 Data Availability

All Davis hydroxyl rotational data described in this manuscript are available through the
Australian Antarctic Data Centre website (under project AAS4157) via the following link
- https://data.aad.gov.au/metadata/records/Davis_OH_airglow. The satellite data used in
this paper were obtained from the Aura/MLS archive at the Goddard Earth Sciences (GES)
Data and Information Services Center (DISC) Distributed Active Archive Center (DAAC)



(see https://disc.gsfc.nasa.gov/ and https://mls.jpl.nasa.gov) and the SABER data archive
(see http://saber.gats-inc.com/data.php) and are publicly available. The ERA5 reanalysis is
publically available from the Copernicus Climate Data Store
(https://climate.copernicus.eu/climate-data-store). ERSST and OI-SST data sets are
publicly available from the NOAA Physical Science Division website
(https://www.esrl.noaa.gov/psd/).

## 688 Author Contribution

WJRF managed data collection, performed data analysis, and prepared the manuscript
and figures with contributions from all co-authors.
ARK analysed Aura/MLS satellite data and provided interpretation and manuscript and
figure editing.
FJM analysed SABER data, and provided interpretation and editing of the manuscript,
figures, and references

## 696 Competing Interests

The authors declare that they have no conflict of interest.

## 699 Acknowledgements

The authors thank the dedicated work of the Davis optical physicists and engineers
over many years in the collection of airglow data and calibration of instruments. We thank
Dr. Damian Murphy for provision of the MFSA radar data from Davis. These projects are
supported by the Australian Antarctic Science program (projects AAS 4157 and AAS



4025).  Aura/MLS data used in this study were acquired as part of the NASA's Earth-Sun
System Division and archived and distributed by the Goddard Earth Sciences (GES) Data
and Information Services Center (DISC) Distributed Active Archive Center (DAAC).
SABER were obtained from http://saber.gats-inc.com/data.php. ECMWF/ERA5 data were
also obtained from the Copernicus Climate Data Store (https://climate.copernicus.eu/
climate-data-store). ERA5, ERSST and OI-SST data sets were accessed via the KNMI
Climate Explorer site (https://climexp.knmi.nl/); ERSST and OI-SST original data from
the NOAA Physical Science Division (https://www.esrl.noaa.gov/psd/). CESM-WACCM
data were obtained from the British Atmospheric Data Center (http://badc.nerc.ac.uk). We
thank those teams and acknowledge the use of these data sets. This work contributes to the
understanding of mesospheric change processes coordinated through the Network for
Detection of Mesospheric Change (https://ndmc.dlr.de/).



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

Figures

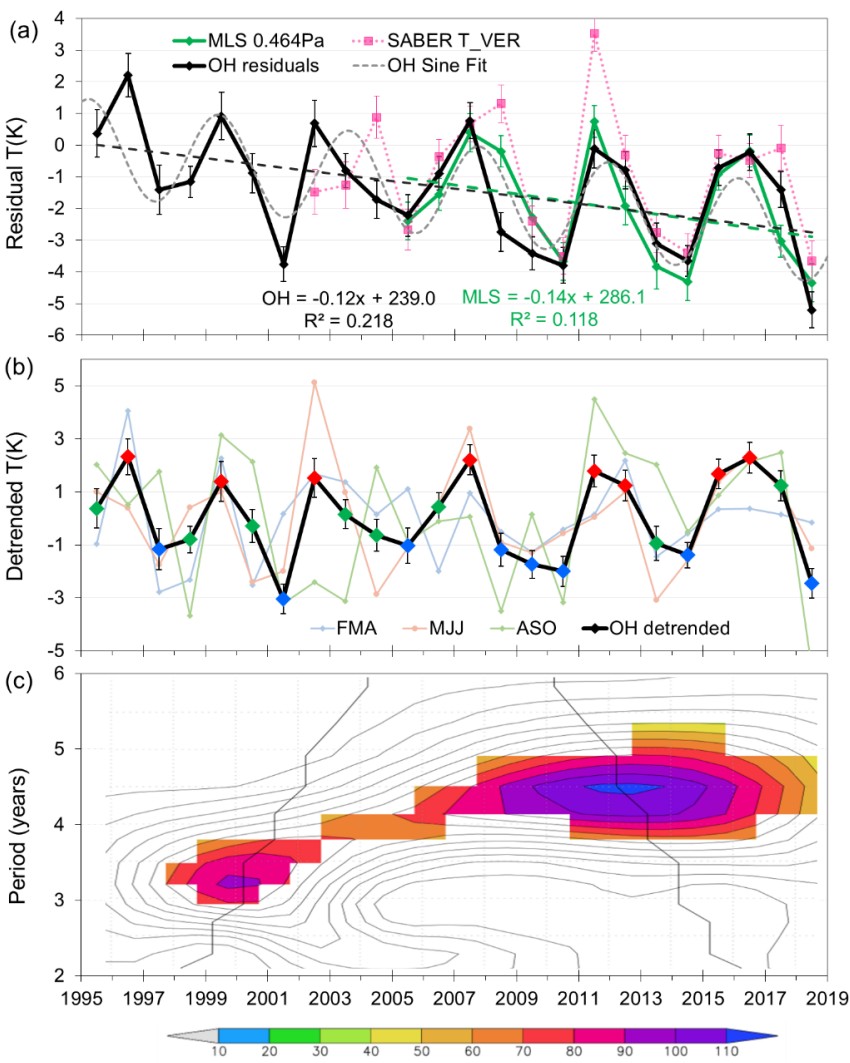

Figure 1. (a) Davis OH winter mean residual (solar response removed) temperatures (black
line, standard error in mean error bars, dashed linear fit) compared with Aura/MLS
[AMJJAS] mean residual temperatures for 0.0046 hPa (green line, standard error-in-mean
error bars, dashed linear fit) and TIMED/SABER (pink dotted line, standard error-in-mean
error bars). Gray dotted line is a sinusoid fit (peak-peak amplitude 3.0 K period 4.2 years).
(b) Detrended Davis OH winter mean temperatures [AMJJAS] (black line, long-term linear
fit removed) compared to FMA, MJJ and ASO monthly averages (red, green and blue
points mark warm, mid and cold years for composite studies). (c) A Mortlet wavelet
transform (order 6) of the detrended Davis OH winter mean temperatures. Coloured
sections are power significant above 90% level as per colour bar. The black line indicates
the cone of influence; points outside have been influenced by the boundaries of the time
series.




Figure 2. (a). Correlation of the Aura/MLS 0.0046 hPa grid box QQO residual temperature
signal at Davis (left hand time series panel) with each grid box of the Aura/MLS 0.0046hP
global temperature field gridded in 5° x 10° bins.  Equi-rectangular and polar projections
of the correlation (R) are shown (hashed areas are significant at the 90% level). Davis
location indicated by green dot. (b) As for (a), but correlation of the 0.0046 hPa SH polar
cap average (65 °S - 85° S; green circle) with each grid box of the Aura/MLS temperature
fields at various pressure levels as indicated.



923

Figure 3. Composites of the ERA5 [AMJJAS] geopotential anomaly, for cold, mid and
warm years of the Davis detrended winter average QQO signal. Pressure levels are
indicated on the right hand colour bar. The colour scales are in m of geopotential height.
Hashed areas on the plots are significant at the 90% level.



928

Figure 4. Composites of the ERA5 [AMJJAS] meridional wind anomaly, for cold, mid and warm years of the Davis detrended winter average QQO signal. Pressure levels are indicated on the right hand colour bar. The colour scales are in m/s. Hashed areas on the plots are significant at the 90% level.

933

934

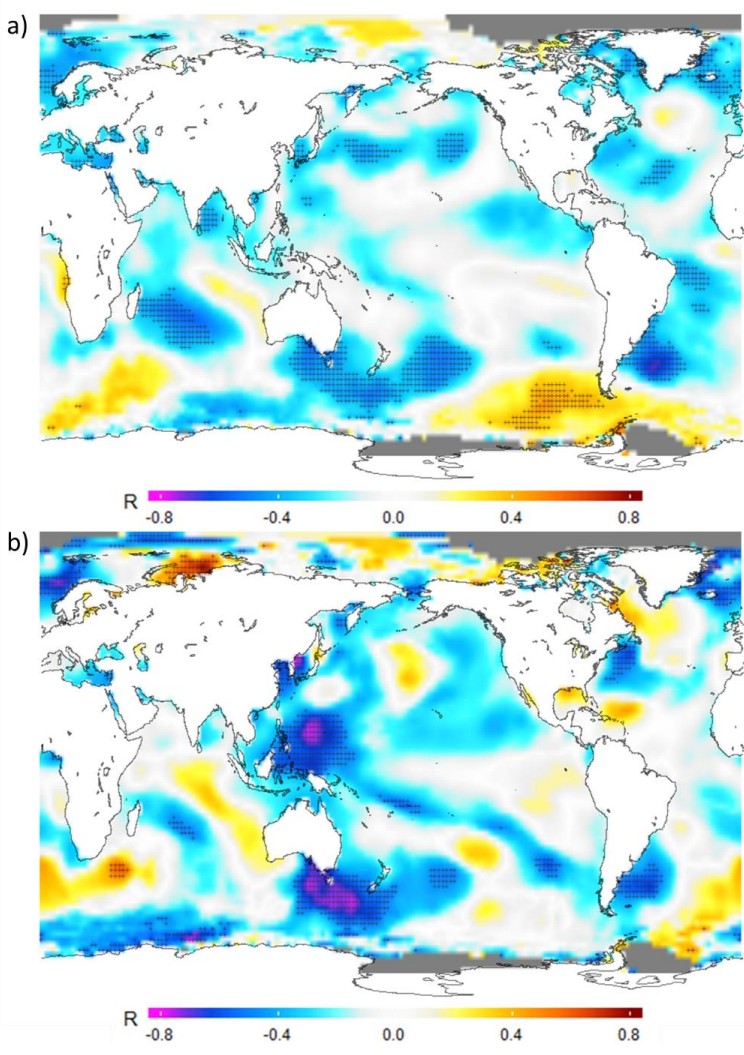

935

Figure 5. Correlation maps of the Davis OH residual (24 year) QQO signal (a) and MLS
0.0046hPa [AMJJAS] polar cap average residual (14 year) QQO signal (b) with the
Extended Range Sea Surface Temperature [AMJJAS] average anomalies. Hatched colours
are significant at the 90% level. Grey areas are permanent sea ice.







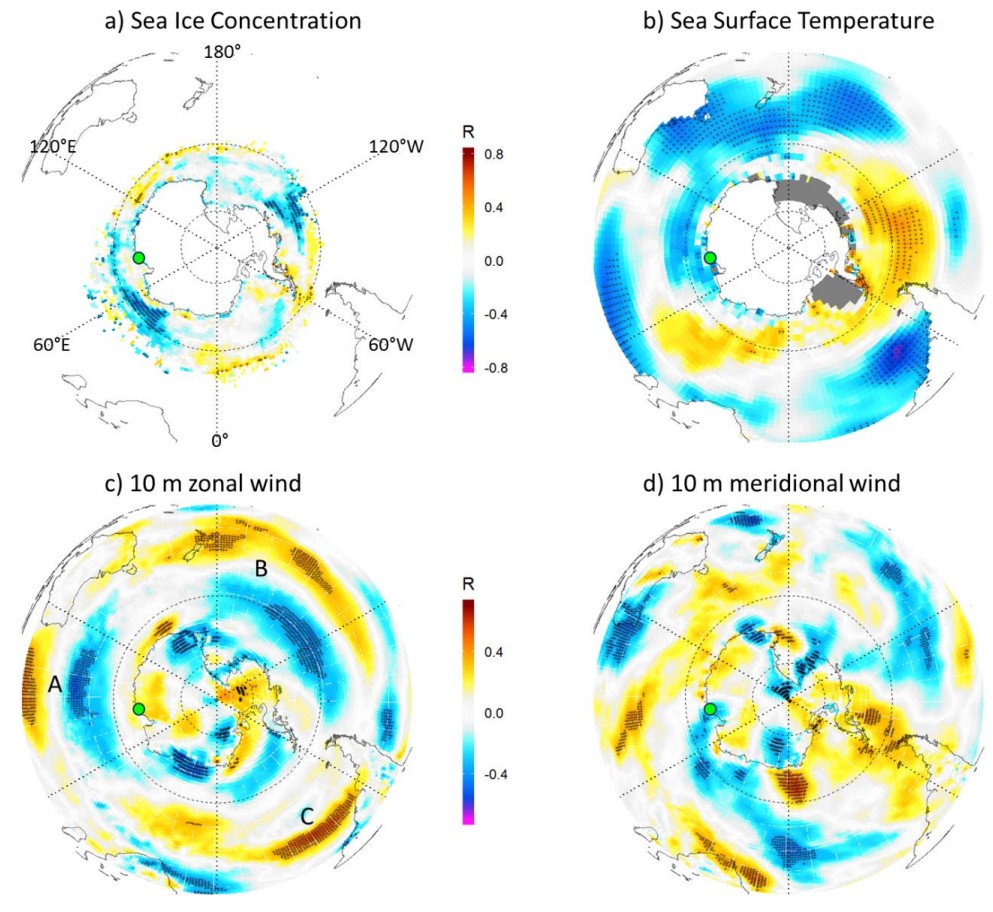


Figure 6. Correlation maps of Davis OH 24 year QQO signal (green dot) with a) Sea Ice
Concentration (OI-SST; Reynolds et al., 2002)  b) Sea surface temperature (ERSSTv5) c)
Surface (10m) Zonal wind anomaly (ERA5) and d) Surface (10m) Meridional wind
anomaly (ERA5). Hashed areas are significant at the 90% level


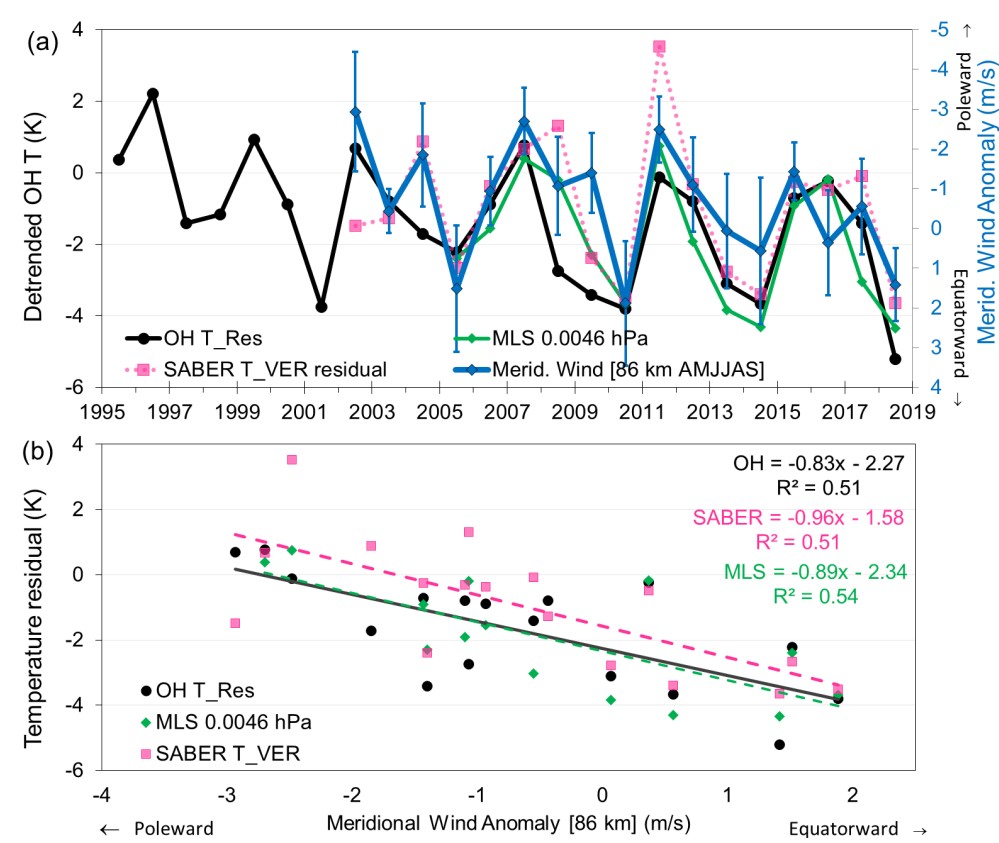

Figure 7. (a) Davis OH, Aura/MLS (0.464 Pa level) and SABER (T_VER) residual
temperatures compared to the AMJJAS mean meridional wind at 86km measured by
MFSA radar at Davis. (b) The relationship between Davis OH, Aura/MLS and SABER
residual temperatures and meridional winds at 86 km above Davis. OH and SABER are fit
over a common era (2002-2018) for comparison. MLS is fit over 2005-2018.






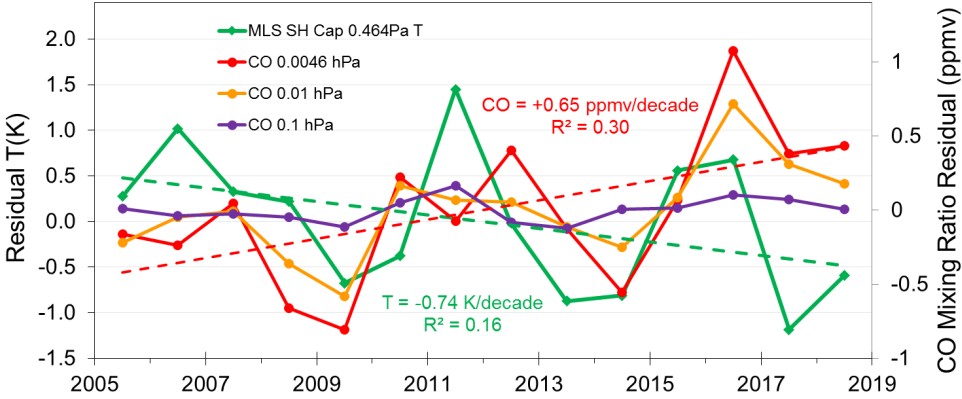


Figure 8. Comparison of Aura/MLS SH polar cap (65° S - 85° S) winter time series of
temperature at 0.0046 hPa (green line and dashed linear fit -0.74 K/decade) with CO mixing
ratio at 0.0046 hPa, 0.01 hPa and 0.1 hPa (red line with dashed fit +0.65 ppmv/decade,
orange line and purple line respectively). The data have been averaged over months
AMJJAS, and had seasonal and solar cycle variations subtracted.







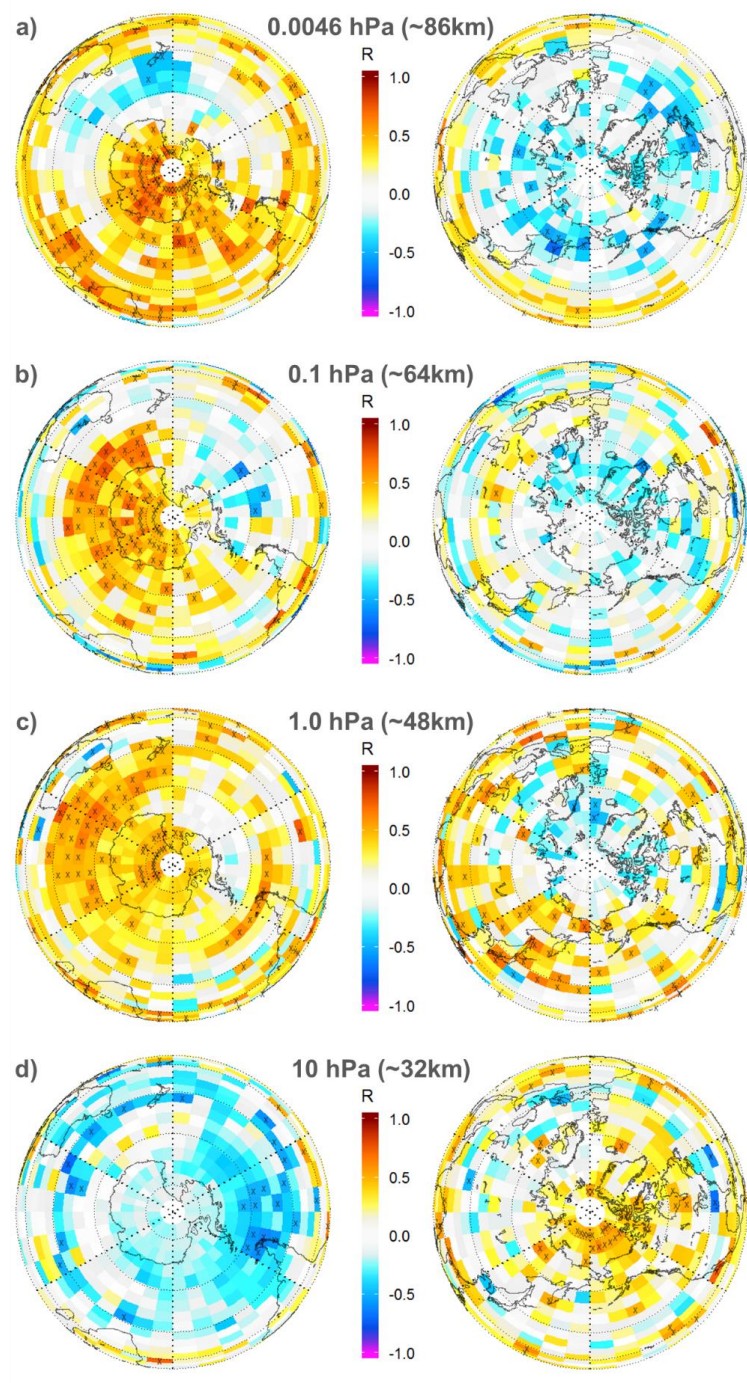


Figure 9. SH and NH projection maps of correlation between the detrended Aura/MLS
0.0046 hPa, SH polar cap (65° S - 85° S), AMJJAS average temperature time series with
residual Aura/MLS CO mixing ratio in each grid box at the pressure levels indicated.
Crosses indicate the correlation is significant at the 90% level.


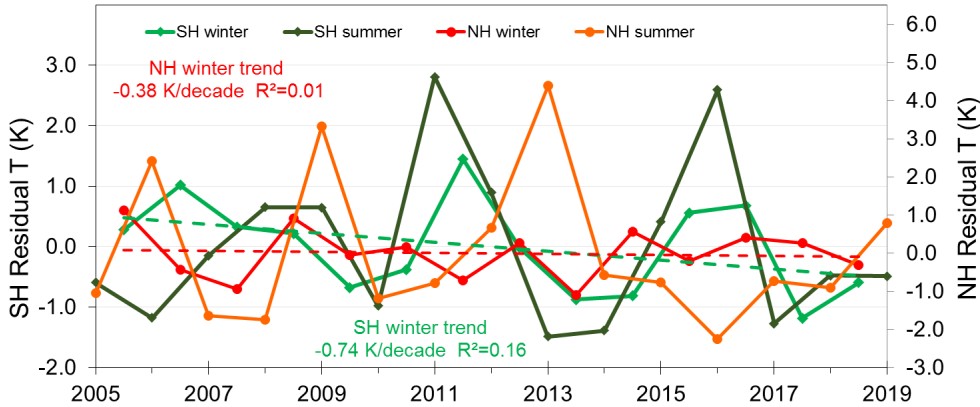

Figure. 10. Time series of Aura/MLS temperature residuals averaged over the SH polar cap
(65° S - 85° S) for winter months (AMJJAS; light green line) and summer months
(ONDJFAM; dark green line), and the NH polar cap (65° N - 85° N) for summer months
(AMJJAS; orange line) and winter months (ONDJFAM; red line).