# Peer review of "Analysis of 24 years of mesopause region OH rotational temperature"

_Atmospheric Chemistry and Physics, 2019_

## Referee Comment (RC1) · Anonymous Referee #1 · 18 Feb 2020

Reviewer Report on the manuscript acp-2019-1097

Analysis of 24 years of mesopause region rotational temperature observations at Davis, Antarctica. Part 2: Evidence of quasi-quadrennial oscillation (QQO) in the polar mesosphere

By W.J.R.French, A.R.Klekociuk, and F.J.Mulligan

[Figure]

General Remarks

1) The paper studies multi-annual variability in the lower and middle atmosphere up to the mesopause which is an interesting objective. 2) Emphasis is on quasi-quadrennial oscillations (QQO) observed in mesopause hydroxyl temperatures in a 24 year time series. 3) The variability signal is also seen in many other parameters as temperature, winds , geopotential, trace gas mixing ratios, SST, sea ice. These are obtained from various sources as satellites (MLS, SABER), ground based radar, ERA 5, etc.. 4) The analysis concerns vertical as well as meridional and zonal structures of the middle atmopsphere. This is very interesting and worth publishing! 5) However: More than half of the paper (pictures) deals with these parameters, only, and not with the Davis temperatures! The title of the paper is, therefore, inappropriate and misleading. It should be changed to something more general, and the manuscript should be rearranged accordingly. I know that this is not an easy task, as the interannual variability of the middle atmosphere is a very extended topic, and the data shown in the paper form only part of it. Nevertheles I recommend rewriting the paper in this direction, rather than turning it down. (To make it clear: I do not recommend that the authors write a review of middle atmosphere interannual variability, but that they state that their work forms an essential part of such a larger overview.) 6) The paper is well written, but many of the figures need improvement. 7) The paper is recommended for publication after major changes have been made.

Major Comments

1) Fig.1: a) The period of 4.2 years is not very convincing! In the years before 2006 the agreement of Davis-T, Saber-T, and the 4.2 yr oscillation curve is marginal! Please give an error bar for the 4.2 yr period value (see for instance Kalicinsky et al., ACP 16, 15033, 2016; Kalicinsky et al., JASTP 178, 7, 2018). b) How did you detrend solar cycle and long-term trend? Simultaneously or in an iteration?

2) Fig. 1 and related text: Figure 10 might be moved to this part of the paper to illustrate

that the interannual variability is fairly different in summer, winter, North and South.

3) Fig.2: a) This is mostly a global analysis, and only a small part is from OH temperatures. Hence, my General Remark #5 applies. This is also the case in Fig.3 and many other places of the paper, especially for most of Section 4. My suggestions in the following assume a paper version in which title and text have been modified already. b) The "hashed areas" are indicated by crosses. These are difficult to discern! This also applies to the following figures, especially if the background colour is blue! The paper would become much more readable if this was improved! c) Fig.2b shows the vertical structure of the interannual variability, which is very interesting. However, the altitude resolution is poor: it only shows that the mesosphere differs from the stratosphere. As described in Section 2.2 there are more altitude levels available. Therefore please complement the left hand column of Fig. 2 by the altitude levels missing. This should show whether the vertical phase distribution is continuous or steplike (as the ones of Offermann et al., 2015).

4) Fig.3 and Fig.4: Please give time series as in the left hand panel in Fig.2.

5) Fig.5: Please give a time series for SST (near to Davis)!

6) Fig. 6: Please give time series!

7) Section 4.1: I understand that the authors are interested in showing a conection between SST and sea ice, and the upper atmosphere QQO. However, Sect.4.1 is not really suitable for this. The correlations discussed are marginal or non-existent (L387: R = -0.09 is not a correlation). The text refers to many literature papers that one would need to read in order to understand the text. Finally, correlation of a parameter below the tropopause with one above the tropopause is generally a delicate business, as is, for instance, indicated by Fig.3, 4. Altogether, a much mor extended analysis would be needed, as the authors state by themselves. As this is beyond the scope of this paper, I recommend to summarize this Section in a few sentences or omit it, at all.

8) Section 4.3, L444pp: Obviously, the data of Dyrland and of Espy are Northern Hemisphere data. How does this compare to your SH results? Can you give a picture?

9) Section 4.4: CO is an important parameter, and its analysis is interesting. However, the correlation $R^2$ = 0.13 at 14 datapoints is barely significant.

10) L483: Do you mean concentrations or mixing ratios in the text and figures?

11) Fig.9a, L493pp: I could detect the "crosses of significance" only if I used a strong magnifying glas.

12) Sect.4.5; Lines 507, 510: If you omit two or four data points from a series of fourteen, the resulting conclusions are very dubious. Please phrase more cautiously!

13) Lines 529, 530: Apparently, WACCM does not detect your QQO, either! Why then show this Section 4.6?

14) L 610-612: This is a misunderstanding: The periods cited are from the Duffin oscillator which is a non-linear oscillator in the ocean. However,the oscillations discussed by Offermann et al., 2015, are intrinsic in the atmosphere! These authors state that their results are not in contradiction to other authors who reported solar cycle harmonics. They note, however, that it is difficult to disentangle these two types (Section 6.2, last paragraph in that paper).

15) Summary, L 626pp: Please state clearly, that the Davis data are winter data, and that summer values are lacking. Fig.10 shows that there may be large differences!

Minor Comments

1) Line 38: relationship is suggesting

2) L 50: French et al., 2020

3) L 80 : including high

4) Fig.3 – 5 : Please indicate location of Davis.

5) Fig.8: Please give error bars.

6) Fig. 10: Please give error bars. Orange and red lines difficult to distinguish!

---

## Referee Comment (RC2) · Anonymous Referee #2 · 3 Mar 2020

Review of "Analysis of 24 years of mesopause region OH rotational temperature 1 observations at Davis, Antarctica. Part 2: Evidence of a quasi-2 quadrennial oscillation (QQO) in the polar mesosphere" by French et al.

This manuscript reports on the detection of a quasi-quadrennial oscillation detected in the SH polar winter OH nightglow, MLS and SABER temperatures. The feature is characterized after the removal of the solar signal and a linear trend as described in Part 1 of the study. The signal has a counterpart in the NH according to MLS data. The authors suggest a relationship between the temperature QQO and the changes in the mesospheric meridional winds and the corresponding changes in downwelling through adiabatic cooling and heating. The authors also show a significant correlation with the SST and provide an explanation of the connection with the mesospheric QQO through a changing GW filtering by modulated PWs.

I found the paper interesting in general and well written, the methodology used is convincing, and the results are new. Therefore, I think it is in principle suitable for publication in ACP. However, I must admit that I sometimes got lost when reading the discussion in Sect. 4. There are several places where it is not clear to me what the authors want to state. There are cases where mechanisms that are clearly not connected to the feature (for example, due to their time-scale) are even described in detail. The discussion should be more concise in order to make the reading more fluent and the reasoning more understandable. It should provide briefer but still comprehensive information only on mechanisms that may potentially be connected to the mesospheric QQO. Also, the authors should try to use a more cautious wording because they only suggest the origin of the feature through reasonable connections but do not provide a solid proof. In summary, in order to improve the paper, I urge the authors to re-write and significantly shorten section 4. Besides taking into account the suggestions listed below, I think the paper would that way be ready for publication in ACP.

Comments:
L19 "peak-to-peak" amplitude
L27 suggesting -> suggests
L27 That the authors have found a correlation between the QQO and the CO vertical transport does not necessarily mean that a "substantial part of the QQO is the result of adiabatic heating and cooling driven by the meridional flow" but just that the QQO is plausibly linked, at least, to the change in meridional flow. Please, provide some quantification proving such substantial contribution or re-write the sentence.
L30 In case it is not confirmed by modelling, please, replace 'suggest a tidal or planetary wave influence in' by 'is consistent with tidal or planetary waves influencing'
L32: 'potentially' -> 'plausibly'
L40, insert comma after Antarctica
L78. Close parenthesis
L147 'Retrieved' might be ambiguous in this context. Do you mean 'selected'?
L148 Did you select both daytime and nighttime MLS profiles? If so, did you check mean biases with respect to nighttime only profiles?
L153 The selection of this altitude is somehow artificial. The OH-layer altitude changes with time. Explain using SABER data the year-to-year change of this altitude and its effect on MLS data. Also explain the relationship between the QQO in temperature and the QQO in altitude.
L184 What is the error in temperature due to a 1km-vertical shift?
L190 Your figure 2b shows that the seasonal variation of the QQO is significant, further, its year-to-year change is quite important. Indeed, according to the colour lines in Fig.1b, the large 2011 residual temperature seems to be influenced by the lack of measurements during doys 141-195, with lower temperatures. This suggests that, due to a potential

sampling bias, comparing SABER data with the ground-based OH dataset shown in Fig.1a, that includes doys 141-195, might be misleading. Also there should be a bias in the derived trend and solar term due to that fact. I suggest to compare just with results from ground-based OH using only the coincident days, or at least FMA+ASO data in Fig. 1b, even if this needs from another figure. This may lead to a better SABER and Davis OH agreement. Indeed, I was expecting a better agreement than with MLS, given that SABER takes into account potential variations of the layer altitude.

L190. I wouldn't say that not observing from 21 May to 14 July means that 'SABER samples the same days'. Please, correct.

L193 What trend and solar terms do OH data provide when not using doys 141-195?

L197. According to a sentence written four lines above this one, the SABER T_VER trend is -0.77K/decade. This sounds contradictory with the -0.13 K/year mentioned in L197.

L198 This anti-correlation has a strong seasonal dependency, being significant in late autumn and very small in early spring for their case (e.g., Garcia-Comas et al., 2017) this. It is therefore not surprising that you get only slight anti-correlation when mixing temperatures for the two intervals doy 75-140 and 196-L208 End sentence with a dot.

L250 Is this MLS temperature also?

L262 What happens to the temperature-altitude anti-correlation if you separate these two intervals? This should also help to answer my previous comment on the effect of OH altitude variation on temperature.

L260 Why the detrended OH temperature is compared with the GPH anomaly instead of the detrended GPH?

L294 Perhaps, this is more like a WN1 + WN2 structure. The WN2 is more clearly seen at 50hPa.

L308 I do not think the feature is a wave-3 structure but more a wave-2. Note that the apparent wave-3 is due to the equirectangular projection.

L347 Do you mean 10m meridional and zonal winds?

L350 I do not see a clear correlation between sea ice and the OH QQO, except for mesoscale areas. The QQO is however more clearly correlated with the SST in Fig.6b or the winds in Figs 6c-6d. That the sea ice concentration should be related to the SST and the winds seems not really big news. Then, what is the additional information from OH-QQO and the sea ice correlation here? If it is difficult to provide a satisfactory answer, I would suggest changing the title of section 4.1 to "SST and winds", suppressing Fig.6a and the long discussion below on the sea ice QQO (L376-L386).

L359 Do you mean cyclonic circulation anomalies? Note that you show correlations in Figure 6 and not winds.

L366 I think the reasoning is in general correct but what the authors show here is a strong connection between the QQO and the SST and also links to the surface circulation. The phenomenon where the origin resides cannot be determined from this type of analysis.

L369 Again, unless I missed something, I do not understand the interest of mentioning the sea ice in this work unless the authors see it as a potential source of their OH QQO. Please, provide a more convincing argument or suppress.

L385 QQO temperature signal at what level?

L405 colder 'mesospheric' temperatures

L414-426 I do not really understand the need for this discussion. Even if the tides are affected by the QBO, there is no impact of QBO-ENSO on OH QQO. Could you please extend on the point here?

L415 Not particularly but only in the cold years.

L423 'longitudinal' wave patterns

Figure 7a. Is that really detrended OH T, as the x-axis title states? Please, write also in the caption "zonal mean meridional wind anomaly". Also, this figure is somehow redundant with Fig. 1. Winds could be overplotted there.

Figure 7b. I think Fig. 7b is not needed. The correlation is clearly seen in Fig. 7a. Remove Fig 7b or, at least, move it to the supplement.

L439 Do you mean the background 'meridional' wind?

L438-442 I think it is well known that the mesospheric poleward circulation is connected to downwelling and adiabatic heating below 100km. I do not think the references credited by the authors are the first ones showing that.

L444 'the adiabatic action of the residual meridional circulation' sounds confusing and may be misinterpreted

L452-L454 Please, remove this sentence. It is not necessary to provide the sources and the sinks of CO. Just knowing that it is a dynamical tracer due to its long lifetime, particularly, during polar winter, is enough.

L464 I find the anti-correlation between temperature and the CO trends interesting. This might be out of the scope of this paper and not worth to mention in the paper but: What are the errors of the CO measurements? Is the instrumental drift characterized? Can the authors provide a link for the trend anti-correlation? Perhaps CO2 increase?

L473 Would the correlation be better if the CO detrended data were used? Given the nature of the link between CO and T QQOs that the authors are providing, does it make sense to provide 8 maps with projections for selected altitudes instead of just one plot with a lat-z cross section? On the other hand, if the QQO is clearly exhibited in the SH polar cap average (see Fig. 2 and Fig 3), why the correlation with T and CO QQO is not ubiquitous south of 65S?

L495 Temperatures at what pressure level?

L498-500 I would not call that "somewhat smaller". There is a very weak QQO in the NH winter.

Section 4.5 Why is this section called "inter-hemispheric coupling"? Certainly, Figure 10 shows NH and SH residual temperatures but their connection is not discussed at all.

L530 If that is the case, why the QQO is largest during the SH summer?

L544 Please, state this is true for SH.

L550-554 I do not think that results from Sato et al. (2012) (showing a close to annular structure around the Arctic edge, with maximum values over the Antarctic Peninsula) are well reproduced by Fig. 2a and 2b (showing more or less homogeneous correlations over the Arctic and a lack of correlation over the Peninsula).

P554-558 Could the authors provide some conclusion after these sentences and make clear the relationship with their work?

L562-570 The vertical and latitudinal structure of Zhang et al.'s TO is similar to the QBO. However, the authors showed no correlation of their QQO with the

QBO. Then, why this discussion of that TO if the four-year oscillation is not even discussed in Zhang et al.? Please, clarify the interest of this discussion or remove. Please, also note that, even if Zhang et al. showed no evidence of the four-year oscillation at 45 km, they did at 85km.

L570-572 From the discussion in the previous section, I thought that the authors were suggesting that mesospheric QQO was related to orographic GWs.

L573-575 Did Liu et al. find any QQO in their GW potential energy at the equator. Figure 2b shows a QQO there at e.g. 81km

L579-585 That a changing eddy diffusion is not needed anymore to explain SABER CO2 trends has nothing to do with its potential link with a QQO.

L594 Replace the fist two dots by a comma

L596 Offermann et al. note in their introduction periods ranging from around 2 yrs to 11 yrs. What periods do the authors refer to in this sentence? All of them?

L623-624 If thepolar cap also shows a QQO of similar amplitude and in phase with Davis (as you write in the next sentence, it is obvious that the Davis QQO is positively correlated with the polar cap QQO. Please, re-write.

L631 extends vertically "at least" from the mesopause

L646 Would a significant variability in atmospheric tides be actually expected at this high polar latitudes?

L653-655 I do not think the authors prove this connection to be "most likely".

L669-672 Is the Davis QQO anti-correlated with the NLC boundary latitude reported by Russell et al.?

Fig 2 caption. I think the caption for b) is not correct. If "as for a)", shouldn't it be just "correlation of the SH polar cap average", without "0.0046hPa"?
L921 What green circle?
Figure 6. Please, indicate the maximum and minimum values in the legend of the color scale for the lower panels. Are the two color scales the same? If the answer is yes, just keep one of them. If the answer is no, use the same color scale for the four panels.
Figure 10. Please, use the same scale for both the NH and the SH residuals. Otherwise, they are not easily compared.

---

## Author Comment (AC1) · 14 May 2020

**Reviewer Report on the manuscript acp-2019-1097**

Analysis of 24 years of mesopause region rotational temperature observations at Davis, Antarctica. Part 2: Evidence of quasi-quadrennial oscillation (QQO) in the polar mesosphere

By W.J.R.French, A.R.Klekociuk, and F.J.Mulligan

**General Remarks**

1) The paper studies multi-annual variability in the lower and middle atmosphere up to the mesopause which is an interesting objective. 2) Emphasis is on quasi-quadrennial oscillations (QQO) observed in mesopause hydroxyl temperatures in a 24 year time series. 3) The variability signal is also seen in many other parameters as temperature, winds, geopotential, trace gas mixing ratios, SST, sea ice. These are obtained from various sources as satellites (MLS, SABER), ground based radar, ERA 5, etc.. 4) The analysis concerns vertical as well as meridional and zonal structures of the middle at-mopsphere. This is very interesting and worth publishing! 5) However: More than half of the paper (pictures) deals with these parameters, only, and not with the Davis temperatures! The title of the paper is, therefore, inappropriate and misleading. It should be changed to something more general, and the manuscript should be rearranged accordingly. I know that this is not an easy task, as the interannual variability of the middle atmosphere is a very extended topic, and the data shown in the paper form only part of it. Nevertheles I recommend rewriting the paper in this direction, rather than turning it down. (To make it clear: I do not recommend that the authors write a review of middle atmosphere interannual variability, but that they state that their work forms an essential part of such a larger overview.) 6) The paper is well written, but many of the figures need improvement. 7) The paper is recommended for publication after major changes have been made.

We thank the reviewer for the comments. As a general response to item 5), thank you for the suggestion regarding the title. This work forms part 2 of a two part series reporting on the long-term measurements of OH rotational temperatures at Davis. In part 1 (acp-2019-1001; "Analysis of 24 years of mesopause region OH rotational temperature observations at Davis, Antarctica. Part 1: Long-term trends.") we focus on the solar cycle response and long term trend components. In this part, we focus on the residual variability observed in those data (the QQO). We decided to separate these sections as they deal with distinctly separate aspects of the long-term measurements.

The principle and foremost observation in both these reports are the trends and variability in the OH rotational temperature data set from Davis. We use many different (publicly available) data-sets in this part to put the Davis observations in global context and use correlation and composite analyses to understand the source and mechanism of the

apparent QQO in Davis OH residual temperatures. We provide evidence of the feature by comparisons with Aura/MLS and SABER temperatures and search for clues to its origin in wind, pressure and sea surface temperature data. We therefore do not think the title is inappropriate or misleading.

**Major Comments**

1) Fig.1: a) The period of 4.2 years is not very convincing! In the years before 2006 the agreement of Davis-T, Saber-T, and the 4.2 yr oscillation curve is marginal! Please give an error bar for the 4.2 yr period value (see for instance Kalicinsky et al., ACP 16, 15033, 2016; Kalicinsky et al., JASTP 178, 7, 2018). b) How did you detrend solar cycle and long-term trend? Simultaneously or in an iteration?

a) The period of 4.2 years is obtained with a simple sinusoid fit to the residuals. The period and error estimate is 4.18 ± 0.10 years. Coefficients and errors for all model fit coefficients are provided below. The curve is provided as a guide. It is clearly not a simple sinusoid of 4.2 years which is why the term *quasi*-quadrennial oscillation is used (in much the same way that the *quasi*-biennial oscillation (QBO) is not strictly a 2-year periodicity).

Formula: y ~ Offset + Amp \* sin(2 \* pi \* (Phase - x)/Period)

Parameters:

|               | Estimate   | Std. Error | t value  | Pr(> t )     |
|---------------|------------|------------|----------|--------------|
| Offset(K)     | 0.02455    | 0.25801    | 0.095    | 0.925137     |
| Amp(K)        | 1.49255    | 0.35868    | 4.161    | 0.000483 *** |
| Phase(year)   | 1994.20367 | 0.35537    | 5611.601 |

The comparison with SABER is limited as we cannot compare the same winter data interval due to SABERs yaw cycle. As described in the text 'only days 106-140 and 196-259 are comparable between SABER and Davis-OH over the winter interval and days 141 - 195 (21 May to 14 July) are excluded. We note that the comparison is not as good an agreement as Aura/MLS but still indicates the presence of a QQO feature.

Another version of the wavelet analysis is shown below with 95% confidence contour in white and the ridge as black points, (cone of influence shaded). The ridge varies between 3.29 and 4.46 years.

 b) The solar-cycle and long-term trends are detrended simultaneously with a multiple linear regression model. This is described in detail in part 1 of this work (acp-2019-1001)

2) Fig. 1 and related text: Figure 10 might be moved to this part of the paper to illustrate that the interannual variability is fairly different in summer, winter, North and South.

Thank you for the suggestion. Figure 1 is specifically the Davis QQO observation. It is the source of our identification of the QQO variation, describes the characteristics of the feature, and provides corroborating evidence of the variation from satellite observations specific to the location of Davis.

Figure 10, on the other hand, is polar cap averages (65-85° North and South) of the MLS 0.00464 hPa pressure level and the summer and winter months (AMJJAS and ONDJFM) in each hemisphere.

Our focus is primarily on the SH and we prefer to keep discussion of the hemispheric and seasonal comparison separate in section 4.5

3) Fig.2: a) This is mostly a global analysis, and only a small part is from OH temperatures. Hence, my General Remark #5 applies. This is also the case in Fig.3 and many other places of the paper, especially for most of Section 4. My suggestions in the following assume a paper version in which title and text have been modified already. b) The "hashed areas" are indicated by crosses. These are difficult to discern! This also applies to the following figures, especially if the background colour is blue! The paper would become much more readable if this was improved! c) Fig.2b shows the vertical structure of the interannual variability, which is very interesting. However, the altitude resolution is poor: it only shows that the mesosphere differs from the stratosphere. As described in Section 2.2 there are more altitude levels available. Therefore please complement the left hand column of Fig. 2 by the altitude levels missing. This should show whether the vertical phase distribution is continuous or steplike (as the ones of Offermann et al., 2015).

a) As stated above, the key result, the new observational data, and the focus of the investigation reported here is to explain the QQO feature observed in the Davis OH

temperatures. We would argue that explanation of the QQO variability in the OH temperatures are the whole reason for the study. We have examined the temporal and spatial extent of the QQO signal with available global data sets to place the observation in context and to attempt to identify its source. We believe this is a reasonable title.

b) We have made appreciable efforts to re-work the figures to improve the hashed/stippled areas which indicate significance. Perhaps this option of applying a border to grid cells that pass the significance criterion improves visibility and clarity? We are happy to defer to editorial and publication recommendations on this.

---

## Author Response (AR2)

Reviewer Report on the manuscript acp-2019-1097

Analysis of 24 years of mesopause region rotational temperature observations at Davis, Antarctica. Part 2: Evidence of quasi-quadrennial oscillation (QQO) in the polar mesosphere

By W.J.R.French, A.R.Klekociuk, and F.J.Mulligan

General Remarks

1) The paper studies multi-annual variability in the lower and middle atmosphere up to the mesopause which is an interesting objective. 2) Emphasis is on quasi-quadrennial oscillations (QQO) observed in mesopause hydroxyl temperatures in a 24 year time series. 3) The variability signal is also seen in many other parameters as temperature, winds , geopotential, trace gas mixing ratios, SST, sea ice. These are obtained from various sources as satellites (MLS, SABER), ground based radar, ERA 5, etc.. 4) The analysis concerns vertical as well as meridional and zonal structures of the middle at- mopsphere. This is very interesting and worth publishing! 5) However: More than half of the paper (pictures) deals with these parameters, only, and not with the Davis temperatures! The title of the paper is, therefore, inappropriate and misleading. It should be changed to something more general, and the manuscript should be rearranged accordingly. I know that this is not an easy task, as the interannual variability of the middle atmosphere is a very extended topic, and the data shown in the paper form only part of it. Nevertheles I recommend rewriting the paper in this direction, rather than turning it down. (To make it clear: I do not recommend that the authors write a review of middle atmosphere interannual variability, but that they state that their work forms an essential part of such a larger overview.) 6) The paper is well written, but many of the figures need improvement. 7) The paper is recommended for publication after major changes have been made.

We thank the reviewer for the comments. As a general response to item 5), thank you for the suggestion regarding the title. This work forms part 2 of a two part series reporting on the long-term measurements of OH rotational temperatures at Davis. In part 1 (acp-2019-1001; "Analysis of 24 years of mesopause region OH rotational temperature observations at Davis, Antarctica. Part 1: Long-term trends.") we focus on the solar cycle response and long term trend components. In this part, we focus on the residual variability observed in those data (the QQO). We decided to separate these sections as they deal with distinctly separate aspects of the long-term measurements..

The principle and foremost observation in both these reports are the trends and variability in the OH rotational temperature data set from Davis. We use many different (publicly available) data-sets in this part to put the Davis observations in global context and use correlation and composite analyses to understand the source and mechanism of the apparent QQO in Davis OH residual temperatures. We provide evidence of the feature by comparisons with Aura/MLS and SABER temperatures and search for clues to its origin in wind, pressure and sea surface temperature data. We therefore do not think the title is inappropriate or misleading.

Major Comments

1) Fig.1: a) The period of 4.2 years is not very convincing! In the years before 2006 the agreement of Davis-T, Saber-T, and the 4.2 yr oscillation curve is marginal! Please give an error bar for the 4.2 yr period value (see for instance Kalicinsky et al., ACP 16, 15033, 2016; Kalicinsky et al., JASTP 178, 7, 2018). b) How did you detrend solar cycle and long-term trend? Simultaneously or in an iteration?

a)  The period of 4.2 years is obtained with a simple sinusoid fit to the residuals. The period and error estimate is 4.18 ± 0.10 years. Coefficients and errors for all model fit coefficients are provided below. The curve is provided as a guide. It is clearly not a simple sinusoid of 4.2 years which is why the term *quasi*-quadrennial oscillation is used (in much the same way that the *quasi*-biennial oscillation (QBO) is not strictly a 2-year periodicity) .

Formula: y ~ Offset + Amp * sin(2 * pi * (Phase - x)/Period)

Parameters:

|  | Estimate | Std. Error | t value | Pr(>\|t\|) |
|---|---|---|---|---|
| Offset(K) | 0.02455 | 0.25801 | 0.095 | 0.925137 |
| Amp(K) | 1.49255 | 0.35868 | 4.161 | 0.000483 *** |
| Phase(year) | 1994.20367 | 0.35537 | 5611.601 | < 2e-16 *** |
| Period(years) | 4.18158 | 0.10287 | 40.647 | < 2e-16 *** |

Signif. codes:  0 '***' 0.001 '**' 0.01 '*' 0.05 '.' 0.1 ' ' 1

Residual standard error: 1.26 on 20 degrees of freedom

[Figure]

The comparison with SABER is limited as we cannot compare the same winter data interval due to SABERs yaw cycle. As described in the text 'only days 106-140 and 196-259 are comparable between SABER and Davis-OH over the winter interval and days 141 - 195 (21 May to 14 July) are excluded. We note that the comparison is not as good an agreement as Aura/MLS but still indicates the presence of a QQO feature.

Another version of the wavelet analysis is shown below with 95% confidence contour in white and the ridge as black points, (cone of influence shaded). The ridge varies between 3.29 and 4.46 years.

[Figure]

b) The solar-cycle and long-term trends are detrended simultaneously with a multiple linear regression model. This is described in detail in part 1 of this work (acp-2019-1001)

2) Fig. 1 and related text: Figure 10 might be moved to this part of the paper to illustrate that the interannual variability is fairly different in summer, winter, North and South.

Thank you for the suggestion. Figure 1 is specifically the Davis QQO observation. It is the source of our identification of the QQO variation, describes the characteristics of the feature, and provides corroborating evidence of the variation from satellite observations specific to the location of Davis.

Figure 10, on the other hand, is polar cap averages (65-85° North and South) of the MLS 0.00464 hPa pressure level and the summer and winter months (AMJJAS and ONDJFM) in each hemisphere.

Our focus is primarily on the SH and we prefer to keep discussion of the hemispheric and seasonal comparison separate in section 4.5

3) Fig.2: a) This is mostly a global analysis, and only a small part is from OH temperatures. Hence, my General Remark #5 applies. This is also the case in Fig.3 and many other places of the paper, especially for most of Section 4. My suggestions in the following assume a paper version in which title and text have been modified already. b) The "hashed areas" are indicated by crosses. These are difficult to discern! This also applies to the following figures, especially if the background colour is blue! The paper would become much more readable if this was improved! c) Fig.2b shows the vertical structure of the interannual variability, which is very interesting. However, the altitude resolution is poor: it only shows that the mesosphere differs from the stratosphere. As described in Section 2.2 there are more altitude levels available. Therefore please complement the left hand column of Fig. 2 by the altitude levels missing. This should show whether the vertical phase distribution is continuous or steplike (as the ones of Offermann et al., 2015).

a) As stated above, the key result, the new observational data, and the focus of the investigation reported here is to explain the QQO feature observed in the Davis OH

temperatures. We would argue that explanation of the QQO variability in the OH temperatures are the whole reason for the study. We have examined the temporal and spatial extent of the QQO signal with available global data sets to place the observation in context and to attempt to identify its source. We believe this is a reasonable title.

b) We have made appreciable efforts to re-work the figures to improve the hashed/stippled areas which indicate significance. Perhaps this option of applying a border to grid cells that pass the significance criterion improves visibility and clarity? We are happy to defer to editorial and publication recommendations on this.

[Figure]

c) Indeed, 55 pressure levels are available in the Aura\MLS data set but we have selected levels which are representative of the stratosphere, stratopause region, mesosphere and mesopause region (see fig right).

Plotted below are the additional time series of Aura/MLS [AMJJAS] polar cap (65-85°S) averages at each of the MLS native pressure levels compared to the Davis OH time series. The y-axis temperature scale is common to all series, but they are offset by the pressure (log scale) indicated by the labels.

These plots show the QQO feature is common through the range 0.0046 - 0.1 hPa, (represented by the 0.0046, 0.01 and 0.1 hPa panels in our figure) then there is a transition between 0.1 and ~1hPa (1hPa is shown) followed by a reasonably consistent pattern to the time series below 1hPa (10hPa is shown). The selected levels of fig 2 thus reasonably encompass the range of variability shown over the polar cap and are representative of the 3 different regimes. This figure is added to the supplementary material as figure 2S.

[Figure]

Full correlation plots for some additional levels are also provided below for your reference, but note that the selected levels do in general capture the correlation patterns throughout the atmosphere profile.

[Figure]

[Figure]

4) Fig.3 and Fig.4: Please give time series as in the left hand panel in Fig.2.

Figures 3 and 4 are *composites* of 5° x 10° (latitude x longitude) or 36 x 36 grid cells corresponding to the cold, mid and warm years of the Davis detrended QQO signal shown in Fig 1(b). As composites the series are accumulated by the Davis detrended temperature, not by time.

We have tried several variants of the significance hatching on these plots to the boxing applied on Fig 2 but this is too heavy with the smaller grid cells in ERA5. We prefer the original although we have modified the point density (see examples below). Again, we will work with editors to optimise this.

[Figure]

[Figure]

5) Fig.5: Please give a time series for SST (near to Davis)!

The correlation analysis in Fig 5a does not show, and we do not claim, a significant correlation of the Davis OH QQO signal with SST in the vicinity of Davis (region C). Instead, we note in the text that the "strongest and most consistent patterns of anti-correlation (QQO warmest for below average SST) for the two epochs occur at mid-latitudes in the south-western Pacific Ocean (to the south of Australia and New Zealand, region D), in the south-western Atlantic Ocean (near the east coast of South America), and in the west-central Indian Ocean (to the west of Madagascar, region B). Significant positive correlation is also seen at mid-latitudes south of Africa (region A), and for the longer-term Davis data set, in the south-eastern Pacific Ocean."

[Figure]

We have plotted below timeseries of SST anomaly compared to the Davis OH residuals and Aura/MLS residuals for the mean within the green bounding boxes A,B,C, and D above.

Correlation coefficients for Davis (Dav) and Aura/MLS (MLS) are provided in the legend text. This figure shows the positive correlation with region A and the negative correlation with region D

[Figure]

This figure shows the negative correlations with regions B and the vicinity of Davis region C. Note that the latter is largely sea-ice during the AMJJAS months.

[Figure]

6) Fig. 6: Please give time series!

Figure 6 has been modified to improve significance plotting and time-series of the two
regions of significant correlation (identified as A and B in figure 6a have been provided
below.

[Figure]

Time series for A and B regions of maximum anticorrelation are provided below (correlation
coefficients provided in the legend). Note the inverse sea ice cover axis.

[Figure]

7) Section 4.1: I understand that the authors are interested in showing a conection between SST and sea ice, and the upper atmosphere QQO. However, Sect.4.1 is not really suitable for this. The correlations discussed are marginal or non-existent (L387: R = -0.09 is not a correlation). The text refers to many literature papers that one would need to read in order to understand the text. Finally, correlation of a parameter below the tropopause with one above the tropopause is generally a delicate business, as is, for instance, indicated by Fig.3, 4. Altogether, a much mor extended analysis would be needed, as the authors state by themselves. As this is beyond the scope of this paper, I recommend to summarize this Section in a few sentences or omit it, at all.

We have moved this section to Section 4.4 to provide better flow in the discussion. We have also added text to better describe the motivation for this section. It is not that we are interested in showing a connection but rather, we note that there are regions in the sea ice zone that are significantly correlated with the Davis QQO variation (regions A and B below) that appear to have a wave-2 structure. We propose that both the upper mesosphere and sea ice may be responding to a common driver and suggest that possibly the meridional winds could drive both observations ie "a persistent northward (southward) flow on one side of a circulation anomaly could increase (decrease) sea ice due to the associated flow of relatively cold (warm) air from higher (lower) latitudes and expansion compaction) of the ice edge.

As shown in Fig 6a, specific regions of significant correlation between the Davis OH temperature record and sea ice concentration occur (regions A and B below). On this map, we have added values for the local maxima and minima of the correlation coefficient where these values are significant. ie region A -0.49 and region B -0.56 (see response to item 6 above) The maximum anti-correlation is -0.61 at 55.5°E, 61.5°S (within region A, marked with purple dot)

[Figure]

[Figure]

The correlation given at L387 the reviewer is referring to (R = -0.09), is the coefficient of the correlation between the time series of sea ice anomaly for a large sector of the Ross Sea region from Parkinson et al. (2019) and the Davis OH residual. We have compared with Parkinson et al (2019) because this reference provides a recent assessment of sea ice trends in various sectors of Antarctica.From the plot above it is clear that the Ross Sea region is not well correlated compared to regions A and B. Instead of the Ross Sea region, we now compare with the Amundsen-Bellingshausen sea region used by Parkinson (2019). The correlation is more negative (R=-0.24) but not significant. We have restated the sentence starting at L386 to better convey that there is no significant correlation for this particular region.

8) Section 4.3, L444pp: Obviously, the data of Dyrland and of Espy are Northern Hemi-sphere data. How does this compare to your SH results? Can you give a picture?

The table below shows a comparison of the Dyrland et al. (2010), and Espy et al. (2003) values to the results obtained here.  As noted in the comment, the data in Dyrland et al. (2010) are from the NH (78 N), but the data in Espy et al. (2003) are from the SH (~68 S). Northward wind is positive; at both poles, poleward flow results in warmer temperatures. Figure 3 in Espy et al. (2003) and Figure 3 in Dryland et al (2010) show correlation plots similar to Figure 7(b) in this work.  A reference to earlier work by Garcia and Solomon (1985) has been added in response to a comment from referee 2.

| Reference | Location | Cofficient of linear regression (temperature to meridional wind) (K/m·s$^{-1}$) (Northward wind is positive; at both poles, poleward flow results in warmer temperatures) | Correlation coefficient |
|---|---|---|---|
| Espy *et al*. (2003) (Fig. 3) | Rothera (67.6°S, 61.8°W) | -0.71 | -0.61 |
| Dyrland *et al*. (2010) (Fig. 3) | Longyearbyen (78°N, 16°E) | +0.50 | 0.71 |
| This work | | | |
| OH(6-2) T_Residual | Davis (68°S, 16°E) | -0.70 ± 0.25 | -0.56 |
| MLS T(0.0464 Pa) Residual | Davis (68°S, 16°E) | -0.89 ± 0.24 | -0.73 |
| SABER T_VER Residual | Davis (68°S, 16°E) | -0.96 ± 0.24 | -0.71 |

9) Section 4.4: CO is an important parameter, and its analysis is interesting. However, the correlation R² = 0.13 at 14 datapoints is barely significant.

The square of the correlation coefficient was incorrectly stated as 0.13; it should be 0.19 (as rounded to 2 significant figures) and thus explaining 19% of the variance.

10) L483: Do you mean concentrations or mixing ratios in the text and figures?

As noted at L473 and the caption to Figure 8 and 9, we use CO mixing ratios, with units in ppmv (parts per million by volume). We have clarified the reference to CO at L483 in terms of mixing ratio.

11) Fig.9a, L493pp: I could detect the "crosses of significance" only if I used a strong magnifying glas.

We have improved Fig 9 so that a magnifying glass is no longer required to detect the crosses of significance. The correlation scale is common and applies to all panels so only one scale is now shown.

[Figure]

Figure 9

12) Sect.4.5; Lines 507, 510: If you omit two or four data points from a series of fourteen, the resulting conclusions are very dubious. Please phrase more cautiously!

Section 4.5 has been substantially modified to *'suggest'* the phase and amplitude relationships between summer and winter NH and SH QQO variation given the data set only spans 14 years. We can only base our interpretation on the data available but believe it is worthwhile examining the SH/NH and winter/summer seasonal comparison. The observations drawn from this comparison that the QQO amplitude appears to be larger in summer than in winter (in both hemispheres), and that the NH (summer) is the opposite phase to the SH we dont believe are overly dubious.

13) Lines 529, 530: Apparently, WACCM does not detect your QQO, either! Why then show this Section 4.6?

As we state in section 4.6 "Our interest here is to see if the model physics produces a QQO response in the mesosphere" and have added "particularly as Offermann et al. (2015) had noted that the CESM-WACCM model showed low-frequency variability in temperatures on 3-6 year timescales over Middle Europe"

If WACCM did show a statistically significant QQO it could be explored and understood through the model mechanism. The fact that it shows a sporadic or not statistically significant response implies a limitation of the model. We have added further text to refer to time-period spectra for each ensemble at three pressure levels - these spectra are shown as Fig. S8 in the Supplementary Material.

14) L 610-612: This is a misunderstanding: The periods cited are from the Duffin oscil- lator which is a non-linear oscillator in the ocean. However,the oscillations discussed by Offermann et al., 2015, are intrinsic in the atmosphere! These authors state that their results are not in contradiction to other authors who reported solar cycle harmon- ics. They note, however, that it is difficult to disentangle these two types (Section 6.2, last paragraph in that paper).

Lines 610-612 refer to the association by Offermann et al (2015) of the periods they found in their data with similar periods found in GLOTI data and the NAO index. The only point we wish to make here is that the QQO found in the Davis data has similarity to the periods reported by Offermann et al (2015). We have rephrased the sentences referring to Offermann et al.(2015) in an attempt to clarify the point.

15) Summary, L 626pp: Please state clearly, that the Davis data are winter data, and that summer values are lacking. Fig.10 shows that there may be large differences!

We have specified 'winter average' throughout the summary and conclusions section to make this point clear. In general, the fact that the Davis data are winter measurements is stated explicitly in the Abstract, Introduction, Data sets (section 2.1) and with reference to the months [AMJJAS] for comparison with Aura/MLS, SABER and other data sets. Section 3 and in the discussion in section 4.

Minor Comments

1) Line 38: relationship is suggesting

Changed to 'suggests'

2) L 50: French et al., 2020

Corrected all occurrences referring to Part 1 of the paper.

3) L 80 : including high removed 'at' as suggested

4) Fig.3 – 5 : Please indicate location of Davis.

Davis location has been added to Figs 3-5

5) Fig.8: Please give error bars.

Error bars have been added to Fig 8

6) Fig. 10: Please give error bars. Orange and red lines difficult to distinguish!

Added error bars, modified trend line colors and separated SH from NH to improve visibility. Corrected NH winter trend value.

New

[Figure]

Old

[Figure]

**Response to RC2**

Review of "Analysis of 24 years of mesopause region OH rotational temperature 1 observations at Davis, Antarctica. Part 2: Evidence of a quasi-2 quadrennial oscillation (QQO) in the polar mesosphere" by French et al.

This manuscript reports on the detection of a quasi-quadrennial oscillation detected in the SH polar winter OH nightglow, MLS and SABER temperatures. The feature is characterized after the removal of the solar signal and a linear trend as described in Part 1 of the study. The signal has a counterpart in the NH according to MLS data. The authors suggest a relationship between the temperature QQO and the changes in the mesospheric meridional winds and the corresponding changes in downwelling through adiabatic cooling and heating. The authors also show a significant correlation with the SST and provide an explanation of the connection with the mesospheric QQO through a changing GW filtering by modulated PWs.

I found the paper interesting in general and well written, the methodology used is convincing, and the results are new. Therefore, I think it is in principle suitable for publication in ACP. However, I must admit that I sometimes got lost when reading the discussion in Sect. 4. There are several places where it is not clear to me what the authors want to state. There are cases where mechanisms that are clearly not connected to the feature (for example, due to their time-scale) are even described in detail. The discussion should be more concise in order to make the reading more fluent and the reasoning more understandable. It should provide briefer but still comprehensive information only on mechanisms that may potentially be connected to the mesospheric QQO. Also, the authors should try to use a more cautious wording because they only suggest the origin of the feature through reasonable connections but do not provide a solid proof. In summary, in order to improve the paper, I urge the authors to re-write and significantly shorten section 4. Besides taking into account the suggestions listed below, I think the paper would that way be ready for publication in ACP.

We thank the Reviewer RC2 for your considered and helpful review. We have revised and re-organised the discussion in section 4 to hopefully improve the flow and comprehension of our study and the discussion. Please find specific responses to your review below.  References to figures and sections are as per the reviewed manuscript, but be aware figure order and sections have been rearranged in the revised manuscript.

Comments:

L19 "peak-to-peak" amplitude

Added as suggested

L27 suggesting -> suggests

Corrected

L27 That the authors have found a correlation between the QQO and the CO vertical transport does not necessarily mean that a "substantial part of the QQO is the result of adiabatic heating and cooling driven by the meridional flow" but just that the QQO is plausibly linked, at least, to the change in meridional flow. Please, provide some quantification proving such substantial contribution or re-write the sentence.

'Plausibly linked' is a better term to describe the relationship. Thank you. We will use that. We have added the correlation coefficients for the OH temperature and radar derived meridional wind at 86 km (from Fig 7) and from the CO correlations in Fig 8. The text in this section has been re-written and now reads …

"We find a significant anti-correlation between the QQO temperature and the meridional wind at 86 km altitude measured by a medium frequency spaced antenna radar at Davis (R2 ~ 0.516; poleward flow associated with warmer temperatures at ~0.83±0.21 K/ms-1). The QQO signal is also marginally correlated with vertical transport as determined from evaluation of carbon monoxide (CO) concentrations in the mesosphere (R2 ~ 0.18 at 0.73±0.45 K/ppbvCO). Together this relationship suggests that the QQO is plausibly linked to adiabatic heating and cooling driven by the meridional flow.

L30 In case it is not confirmed by modelling, please, replace 'suggest a tidal or planetary wave influence in' by 'is consistent with tidal or planetary waves influencing'

Modified sentence as suggested.

L32: 'potentially' -> 'plausibly'

Modified as suggested.

L40, insert comma after Antarctica

Inserted comma.

L78. Close parenthesis

Parenthesis closed.

L147 'Retrieved' might be ambiguous in this context. Do you mean 'selected'?

Changed wording to 'selected'.

L148 Did you select both daytime and nighttime MLS profiles? If so, did you check mean biases with respect to nighttime only profiles?

Yes, both daytime and nighttime profiles were selected as there are only ~2 profiles per day, however for our high latitude winter comparison the vast majority of the samples are night-time profiles. The latest version of the Aura/MLS data quality document (https://mls.jpl.nasa.gov/data/v4-2_data_quality_document.pdf) does not raise temperature biases between the ascending and descending passes as an issue over the recommended pressure level range.

There is a small diurnal temperature variation in the upper atmosphere (largely ozone heating). We have measured the diurnal variation to be <2K in the Davis OH temperatures.

Xu et al., 2010 (Jiyao Xu, A. K. Smith, Guoying Jiang and Wei Yuan. "Seasonal variation of the Hough modes of the diurnal component of ozone heating evaluated from Aura Microwave Limb Sounder observations" JGR, VOL. 115, D10110, doi:10.1029/2009JD013179, 2010) report on the diurnal variation in MLS temperatures due to ozone heating throughout the atmosphere which shows a minimum in the mesopause region (their Fig 2.) with the comment that "at 90 km, not all of the absorbed solar energy immediately appears as heat in the atmosphere. Substantial portions of the incident solar energy are stored as chemical potential energy (i.e., as photodissociation products or other reactant species) and released as heat during exothermic reactions, which can take place during day or night."

L153 The selection of this altitude is somehow artificial. The OH-layer altitude changes with time. Explain using SABER data the year-to-year change of this altitude and its effect on MLS data. Also explain the relationship between the QQO in temperature and the QQO in altitude.

The selection of the 0.00464hPa native MLS pressure level as an appropriate representation of an OH equivalent temperature for comparison with the Davis measurements is detailed in Part 1 of this study. This was also a reviewer concern in Part 1 and the following is paraphrased from the response provided.

In selecting the 0.0046hPa level we compared the Davis OH winter average anomaly with MLS [AMJJAS average] anomaly over a range of altitude and pressure levels. (see plots below, the first are altitude ranges, the second pressure levels, both temperatures and the anomaly are shown).

[Figure]

[Figure]

Altitude ranges 78-83km, 83-88km and 85-90km and pressure ranges 0.00215hPa and 0.00464hPa are all in reasonable agreement (<5K in absolute terms) with the OH temperatures, but we know there are biases with MLS (see French and Mulligan, 2010), and we know that the Davis OH temperatures are ~2 K high using LWR transition probabilities compared to those computed with the experimentally measured transition probability ratios determined in French et al., 2000.

We have previously reported, and routinely compare our measured OH temperatures with both Aura/MLS and SABER profiles. In particular, French and Mulligan, 2010 examined biases between Davis OH temperatures and both Aura/MLS and SABER. A significant limitation of SABER for comparisons with Davis observations is the yaw cycle sampling of the satellite. Comparable observations over Davis are confined to two intervals (day-of-year 75–140 and 196–262) and days prior to 106 and after 259 are outside the OH winter averaging interval. Therefore only days 106-140 and 195-259 are comparable and a large part of the winter months is not sampled. As a consequence SABER winter averages do not fit the OH observations at Davis as well as MLS.

In any case for this study, these biases are removed by comparing anomalies. We calculate the Chi-Square goodness of fit parameter between the OH winter average anomaly with the Aura/MLS anomalies. The 0.0046hPa pressure level yields the smallest chi-sqr (14.8) compared to a layer centred on the traditional 87km altitude level (85-90km chisqr=18.8).

This difference is small, but we prefer the pressure level comparison because we agree that the geometrical altitude of the layer varies. Since the hydroxyl layer position is primarily controlled by collisional quenching with $O_2$ and $N_2$ on the bottom-side of the layer, and reaction with atomic oxygen on the top-side of the layer it is the concentration (density) of the reacting species that governs the layer position. Therefore it is reasonable to compare with MLS pressure (proportional to density) levels than on geometrical altitude levels.

With regard to the change of altitude determined by SABER. We have examined the variation in OH layer height over this period in Part 1 of this study (acp-2019-1001; Figure 7 and related text). The overall trend is not significant, the mean winter VER peak altitude is 85.6 km and the standard deviation is 0.32 km, bearing in mind that the OH temperatures are integrated over the 8km FWHM of the layer shape, and for Aura/MLS the FWHM of the averaging kernel at this height is approximately 15 km.

The Figure below shows a comparison of SABER VER (altitude of peak) and corresponding pressure value (mb) for the years 2002-2018 (day 106 – 259 of each year) at Davis Station. The OH peak occurs at pressures in the range 0.00255 hPa to 0.003 hPa ($2.55\times10^{-3}$ - $3.0\times10^{-3}$ Pa). This value lies between two of the Aura MLS levels (0.00464 hPa and 0.00215 hPa) on which the averaging kernels are centered and is in excellent agreement with the Davis MLS 4.2 Temperature PRES plot (page 4 above). An inverse relationship between altitude and pressure at the OH peak is clearly evident, and justifies the selection of a pressure level comparison for OH temperatures over an altitude level.

[Figure]

In a global context, we can also examine the altitude (or pressure) of OH peak as a function of latitude, the figure below shows the variation of the altitude of the OH peak as a function of latitude and day-of-year for the year 2005 from SABER data. The overall pattern shown here is repeated year after year with only minor changes in detail.

[Figure]

Based on the two figures above, the MLS averaging kernel centered on 0.00464 hPa would appear to be a good representative for the temperature of the OH layer.

The relationship between pressure and geopotential height (GPH) is examined below using the MLS data set. Global decreases in GPH anomaly (between ~110 to 240 metres/decade) at the 0.0046 hPa pressure level are consistent with a contraction of the underlying atmosphere and also consistent with the SABER trend in OH mean winter layer altitude for Davis shown in Fig 7. (200metres/decade) and discussed in the text in part 1 of this study.

MLS slope of fit to residual geopotential height at 0.0046 hPa

[Figure]

L184 What is the error in temperature due to a 1km-vertical shift?

The precise answer to this question depends on the vertical temperature profile and the OH volume emission rate vertical profile at the time of the measurement. (The OH-equivalent temperature (T_VER) is determined from a SABER temperature profile weighted with the volume emission rate profile (VER) measured simultaneously.) Using a typical temperature profile at Davis in the middle of the observing season, a 1 km -vertical shift in the OH-layer at 87 km would result in a temperature error of less than 1 K (added to the manuscript). This study uses long-term averages however and we note from above that the standard deviation in emission peak altitude for mean winter profiles is ~320 metres (over 16 years).

L190 Your figure 2b shows that the seasonal variation of the QQO is significant, further, its year-to-year change is quite important. Indeed, according to the colour lines in Fig.1b, the large 2011 residual temperature seems to be influenced by the lack of measurements during doys 141-195, with lower temperatures. This suggests that, due to a potential sampling bias, comparing SABER data with the ground-based OH dataset shown in Fig.1a, that includes doys 141-195, might be misleading. Also there should be a bias in the derived trend and solar term due to that fact. I suggest to compare just with results from ground-based OH using only the coincident days, or at least FMA+ASO data in Fig. 1b, even if this needs from another figure. This may lead to a better SABER and Davis OH agreement. Indeed, I was expecting a better agreement than with MLS, given that SABER takes into account potential variations of the layer altitude.

Figure 2b) does not show any seasonal variation in the QQO. This is the MLS polar cap average for the months AMJJAS at 0.00464hPa. We assume you are actually referring to figure 1b).

Figure 1b) shows the variation in detrended Davis OH temperatures for the intervals FMA, MJJ and ASO compared to the winter mean [AMJJAS]. This is *not* SABER data which has the missing days 141-195 due to the yaw cycle *every* year (not just 2011) .

The comparison with SABER data in Fig 1a) is provided only to demonstrate that the QQO pattern exists in the SABER OH-equivalent temperatures in the vicinity of Davis. We have stated that the SABER data exclude days 141-195 from the winter averaging window (*every year*; L197) and so do not believe the comparison is misleading.

The derived solar response and long-term trend from SABER OH equivalent temperatures in the vicinity of Davis is reported and discussed in part 1 of this study (acp-2019-1001). We have also previously compared Davis-OH and SABER temperatures in French and Mulligan (2010). The focus of this manuscript is to explore the QQO feature and the SABER temperatures provided in Fig 1a serve only to support the existence of a QQO pattern in winter average temperatures. We are aware that the yaw cycle data gap in SABER observations may yield a bias in the direct comparison with OH temperatures, but this is not a study of satellite biases.

L190. I wouldn't say that not observing from 21 May to 14 July means that 'SABER samples the same days'. Please, correct.

Modified sentence to read "Thus only days 106-140 and 196-259 are comparable between SABER and Davis-OH over the winter interval and days 141 - 195 (21 May to 14 July) are excluded."

L193 What trend and solar terms do OH data provide when not using doys 141-195?

The terms are :

Davis OH (1995-2018)
L = -1.68±0.68 K/decade   (-0.26 > L > -3.09 K/decade 95% confidence)
S = 4.11±1.35 K/100sfu  (6.9 > S > 1.31 K/100sfu 95% confidence)
if days 141-195 are omitted from each year (1995-2018) and

Davis OH (2002-2018)
L = -0.912±1.16 K/decade   (1.58 > L > -3.40 K/decade 95% confidence)
S = 5.40±3.03 K/100sfu  (9.55 > S > 1.25 K/100sfu 95% confidence)
if days 141-195 are omitted from each year (2002-2018) to compare with the SABER trends in the vicinity of Davis.

SABER OH-equivalent T_VER (2002-2018)
L = -0.76±1.06 K/decade   (1.52 > L > -3.03 K/decade 95% confidence)
S = 3.38±1.87 K/100sfu  (7.39 > S > -0.63 K/100sfu 95% confidence)
for SABER T_VER within 500km of Davis (given yaw cycle excludes days 141-195)

This comparison is of interest for the trend assessment in Part 1 of this study but is not particularly relevant to the discussion of the QQO in this part.

L197. According to a sentence written four lines above this one, the SABER T_VER trend is -0.77K/decade. This sounds contradictory with the -0.13 K/year mentioned in L197.

Line 194 is the result of the solar cycle and linear trend model fit, whereas line 197 is the trend in the raw temperatures (i,.e., no solar component removed) in order to compare directly with the trend in the VER peak height. Edited the sentence to make this clear.

L198 This anti-correlation has a strong seasonal dependency, being significant in late autumn and very small in early spring for their case (e.g., Garcia-Comas et al., 2017) this. It is therefore not surprising that you get only slight anti-correlation when mixing temperatures for the two intervals doy 75-140 and 196-

See response to L262 below and Figure of the anti-correlation for days 106-140 plus days 196-259 (top plot),  days 106-140 only (middle plot) and days 196-259 only (lower plot).  Days prior to day 106, and after day 259 are not considered because they are outside the observing season at Davis station.

L208 End sentence with a dot.

corrected

L250 Is this MLS temperature also?

Yes, this correlates the Aura/MLS polar cap mean temperature (indicated by the green ring in Fig 2b) with each MLS grid box, on several pressure levels. The sentence has been modified slightly to explicitly state this.

L262 What happens to the temperature-altitude anti-correlation if you separate these two intervals? This should also help to answer my previous comment on the effect of OH altitude variation on temperature.

The temperature-altitude anti-correlation for SABER DoY 106-140 is -4.6 ± 1.4 K/km and is significant at the 95% level (R2 = 0.43), whereas the corresponding value for SABER DoY 196-259 is -3.2 ± 1.3 K/km and is also significant at the 95% level (R2 = 0.29). In view of the difference between the two intervals, we calculated the solar and linear terms as in the case of the Davis OH data.

| | Summary of trends from SABER T_OH-equivalent (2002-2018) at Davis | | | | | |
|---|---|---|---|---|---|---|
| | DOY 106-140, 196-259 | Significant at 95% | DOY 106-140 only | Significant at 95% | DOY 196-259 only | Significant at 95% |
| Solar term, S | 3.4 ± 1.8 K/100 sfu | No | 6.2 ± 2.3 K/100 sfu | Yes | 1.2 ± 2.3 K/100 sfu | No |
| Linear trend, L | -0.77 ± 1.05 K/decade | No | -0.9 ± 1.3 K/decade | No | -0.1 ± 1.3 K/decade | No |

Again, this comparison is of interest for the trend assessment in Part 1 of this study but is not particularly relevant to the discussion of the QQO in this part.

[Figure]

L260 Why the detrended OH temperature is compared with the GPH anomaly instead of the detrended GPH?

These are GPH composites, binned by the detrended Davis OH temperature into warm, mid and cold years, not time correlations. ie the detrended temperatures are only used to define the composite bins. The linear trend in the GPH does not affect the wave pattern observed in the results.

L294 Perhaps, this is more like a WN1 + WN2 structure. The WN2 is more clearly seen at 50hPa.

Agreed, it certainly appears as a WN2 pattern in the warm years meridional wind but more WN1 like in the cold and intermediate years. Modified this sentence to make this point.

[Figure]

L308 I do not think the feature is a wave-3 structure but more a wave-2. Note that the apparent wave-3 is due to the equirectangular projection.

Agreed, zonal wind at 10 hPa is not a wave 3. Corrected to wave-2.

[Figure]

L347 Do you mean 10m meridional and zonal winds?

Yes 10 m or 'near-surface' - as described in that sentence.

L350 I do not see a clear correlation between sea ice and the OH QQO, except for mesoscale areas. The QQO is however more clearly correlated with the SST in Fig.6b or the winds in Figs 6c-6d. That the sea ice concentration should be related to the SST and the winds seems not really big news. Then, what is the additional information from OH-QQO and the sea ice correlation here? If it is difficult to provide a satisfactory answer, I would suggest changing the title of section 4.1 to "SST and winds", suppressing Fig.6a and the long discussion below on the sea ice QQO (L376-L386).

We have added time series for the correlation between the regions of maximum anti-correlation between Davis-OH and the sea-ice cover to the supplementary material (see figures below). Region A has a correlation coefficient -0.49 with Davis and -0.46 with MLS. Region B has a correlation coefficient -0.56 with Davis and -0.65 with MLS.

We do not see sea ice cover as a potential source of the mesospheric QQO. "Correlation does not imply causation". Rather, we note that there are regions that are significantly correlated with the Davis QQO variation (regions A and B below) that appear to have a wave-2 structure and suggest that possibly the meridional winds could drive both observations ie "a persistent northward (southward) flow on one side of a circulation anomaly could increase (decrease) sea ice due to the associated flow of relatively cold (warm) air from higher (lower) latitudes and expansion compaction) of the ice edge."

This section on correlations with sea ice cover has been revised, suppressed, moved from section 4.1 to 4.4 and renamed "Relationship with SST and Antarctic sea ice".

[Figure]

[Figure]

L359 Do you mean cyclonic circulation anomalies? Note that you show correlations in Figure 6 and not winds.

Inserted 'anomalies'

L366 I think the reasoning is in general correct but what the authors show here is a strong connection between the QQO and the SST and also links to the surface circulation. The phenomenon where the origin resides cannot be determined from this type of analysis.

Agreed, we only suggest "this could hint as to the origin" and provide a plausible possibility to explore with other analyses.

L369 Again, unless I missed something, I do not understand the interest of mentioning the sea ice in this work unless the authors see it as a potential source of their OH QQO. Please, provide a more convincing argument or suppress.

See response to L350 above.

L385 QQO temperature signal at what level?

Generally speaking the QQO in the upper mesosphere. We have shown the QQO detected at Davis is correlated with the SH polar cap between 0.0046 hPa, down to 0.1hPa. We have added "in the upper mesosphere" to the sentence to clarify.

L405 colder 'mesospheric' temperatures

Inserted 'mesospheric'.

L414-426 I do not really understand the need for this discussion. Even if the tides are affected by the QBO, there is no impact of QBO-ENSO on OH QQO. Could you please extend on the point here?

We find distinctly different wave patterns in the ERA5 geopotential and meridional wind anomalies during warm and cold years of the QQO so some factor is modifying the behaviour of these parameters on a QQO timescale. Is this possibly tides or planetary waves ?. This discussion makes the point that interaction between tides with the QBO and ENSO can have an impact at inter-annual timescales as reported by Baldwin et al (2019) and Liu (2016). Is a similar interaction occurring to produce these different wave patterns on a QQO timescale?. The discussion merely poses this as a possibility by corollary with the QBO and ENSO interaction.

L415 Not particularly but only in the cold years.

We see a SH wave-1 structure in the composite cold years, but also possible in the mid years. However, have removed the word 'particularly'

[Figure]

L423 'longitudinal' wave patterns

Inserted 'longitudinal'.

Figure 7a. Is that really detrended OH T, as the x-axis title states? Please, write also in the caption "zonal mean meridional wind anomaly". Also, this figure is somehow redundant with Fig. 1. Winds could be overplotted there.

No, they are residual temperatures as stated in the caption. Thank you for picking that up, the y-axis title has been corrected. The meridional wind anomaly is that measured at Davis it is *not* a zonal mean. Thanks for the suggestion, we agree the figure is somewhat replicated from figure 1 but with the addition of the mean meridional winds. The winds could be added as a separate panel in fig 1 but we prefer to keep this separated as the radar winds are not introduced until the discussion in section 4.2.

[Figure]

Figure 7b. I think Fig. 7b is not needed. The correlation is clearly seen in Fig. 7a. Remove Fig 7b or, at least, move it to the supplement.

Figure 7b provides the regression coefficients and $R^2$ values for the meridional wind to OH, Aura/MLS and SABER temperatures referred to in the text. As the relationship of the OH temperature QQO to the meridional wind has the most significant correlation ($R^2$ 0.51) of the factors compared this is a key finding in the investigation and we prefer to keep this figure (time-series and slope of the relationship) in the manuscript.

L439 Do you mean the background 'meridional' wind?

Inserted 'meridional'.

L438-442 I think it is well known that the mesospheric poleward circulation is connected to downwelling and adiabatic heating below 100km. I do not think the references credited by the authors are the first ones showing that.

We agree with the comment. These references are selected because they are among the first to report observing the effect in OH temperatures, which are the data in which the QQO was identified initially in this study. We have added a reference to the earlier work of Garcia and Solomon (1985).

L444 'the adiabatic action of the residual meridional circulation' sounds confusing and may be misinterpreted

The sentence has been modified to avoid possible confusion. "and the long-term linear trend is due to  meridional circulation."

L452-L454 Please, remove this sentence. It is not necessary to provide the sources and the sinks of CO. Just knowing that it is a dynamical tracer due to its long lifetime, particularly, during polar winter, is enough.

The sentence has been removed.

L464 I find the anti-correlation between temperature and the CO trends interesting. This might be out of the scope of this paper and not worth to mention in the paper but: What are the errors of the CO measurements? Is the instrumental drift characterized? Can the authors provide a link for the trend anti-correlation? Perhaps CO2 increase?

Uncertainties in the CO measurements are of the order of 0.3-0.4 ppmv at 0.00464hPa and 0.05-0.07 ppmv at 0.1 hPa. Error bars have been added to figure 8 as shown below.

[Figure]

Instrumental drift and calibration is addressed by the Aur/MLS science team (https://mls.jpl.nasa.gov/products/co_product.php) but is beyond the scope of this work. There are both dynamical and chemical factors that could contribute to the trend in CO. Perhaps there are trends in the large scale upwelling and downwelling via meridional flow that occur over the polar caps and, as you suggest perhaps the trend is associated with $CO_2$ increases. This deserves further study, but is outside of this manuscript aims.

L473 Would the correlation be better if the CO detrended data were used? Given the nature of the link between CO and T QQOs that the authors are providing, does it make sense to provide 8 maps with projections for selected altitudes instead of just one plot with a lat-z cross section? On the other hand, if the QQO is clearly exhibited in the SH polar cap average (see Fig. 2 and Fig 3), why the correlation with T and CO QQO is not ubiquitous south of 65S?

The correlation coefficient with detrended CO and Temperatures is described on L463. We have considered a Lat vs Z plot but this necessarily requires zonal averages whereas Fig 9 shows that the patterns are zonally asymmetric. These levels and the plot format are provided so that they can be compared directly to the plots in Fig 2,3 and 4 (mesosphere levels). With regard to the polar cap we describe in the text "temperature and CO are significantly positively correlated over most of

Antarctica, which is consistent with Fig. 8 and our hypothesis that the QQO temperature variation is an adiabatic response (i.e. increased (decreased) temperature is associated with increased (decreased) CO concentration due to descent (ascent)). This general positive correlation over Antarctica is also seen for CO at 0.1 hPa (Fig. 9b) and is apparent though less clear for CO at 1 hPa (Fig. 9c)."

L495 Temperatures at what pressure level?

inserted "(the 0.0046hPa pressure level)"

L498-500 I would not call that "somewhat smaller". There is a very weak QQO in the NH winter.

We have modified this section substantially and the sentence now reads "we see little evidence of a QQO variation in the NH winter (blue line with linear fit)" referring to Fig 10 (below)

[Figure]

Section 4.5 Why is this section called "inter-hemispheric coupling"? Certainly, Figure 10 shows NH and SH residual temperatures but their connection is not discussed at all.

We have modified the section heading to Hemispheric comparison as your point is correct.

L530 If that is the case, why the QQO is largest during the SH summer?

We do not have measurements over the summer to confirm the Aura/MLS observation of a larger QQO in the summer months [ONDJFM] in Fig 10. We do state "We have attempted to examine the seasonal variability of the QQO signal by dividing averages into intervals FMA, MJJ, ASO (also plotted in Fig 1b). While these shorter term averages obviously suffer from greater uncertainty, there is a suggestion that the QQO is strongest over the winter months MJJ, mid-range in ASO and less apparent in the FMA interval."

Davis is not near the GW hot spot of the Antarctic Peninsula and we also comment from L543 "Gravity wave energy is generally weak in summer but in winter, gravity waves have large amplitudes and are distributed around the polar vortex in the upper stratosphere and mesosphere. The wave energy is not zonally uniformly distributed but is concentrated on the leeward side of the Southern Andes and Antarctic Peninsula. Energy propagation extends several thousand kilometres eastwards which explains the gravity wave distribution around the polar vortex in winter."

L544 Please, state this is true for SH.

That this refers to the SH is stated in the preceding sentence on L543.

L550-554 I do not think that results from Sato et al. (2012) (showing a close to annular structure around the Arctic edge, with maximum values over the Antarctic Peninsula) are well reproduced by Fig. 2a and 2b (showing more or less homogeneous correlations over the Arctic and a lack of correlation over the Peninsula).

The polar projection plots of the SH at 0.01 hPa (not 0.1 hPa as stated in the original manuscript) and 0.0046 hPa shown in Figure 2(c) in this work show a striking resemblance to the pattern in the SH winter months shown in Sato et al. (2012).

Compare the polar projection plots of June, July, August and September from Figure 2 of Sato et al. (2012) shown below with Figure 2(b) at 0.01 hPa and above from this work (also shown below). Please note that the Sato et al. (2012) plots have longitude 0° at the top of each projection plot, while the SH in Figure 2 of this work have 180° at the top of each projection plot. (This is corrected in the revised manuscript)

[Figure]

FIG. 2. Polar stereo projection maps of gravity wave potential energy (shading) in the SH for each month at 10 hPa. Monthly mean zonal winds are shown by thick contours at an interval of 40 m s⁻¹.

[Figure]

P554-558 Could the authors provide some conclusion after these sentences and make clear the relationship with their work?

A concluding sentence has been added to the text after these sentences. "The results of Sato et al. (2012), together with the focussing of GWs into the polar night wind reported by Wright et al. (2016, 2017) suggests GWs as a plausible explanation for the asymmetry in the polar projection plots of the QQO correlation coefficient at 0.01 hPa and above. "

L562-570 The vertical and latitudinal structure of Zhang et al.'s TO is similar to the QBO. However, the authors showed no correlation of their QQO with the QBO. Then, why this discussion of that TO if the four-year oscillation is not even discussed in Zhang et al.? Please, clarify the interest of this discussion or remove. Please, also note that, even if Zhang et al. showed no evidence of the four-year oscillation at 45 km, they did at 85km.

As the referee has noted, Zhang et al. (2017) found evidence of a four year oscillation at 85 km in their analysis.  This is stated explicitly in the second sentence of the paragraph, which is the reason that we include it in our discussion.  Since we have found evidence of the QQO at different altitudes, we note that Zhang et al. (2017) also found a four-year oscillation at 25 km.  Zhang et al. (2017) do not discuss the 4-year oscillation (as we state), presumably because it has a much lower amplitude than the periods on which they focus.  We make mention of the three-year oscillation simply because Zhang et al. (2017) suggest that it may arise from a modulation of the QBO by the semiannual oscillation.  This struck us as interesting, and we included it here for that reason.  We examined our data for evidence of a similar explanation for the QQO, but we did not find any evidence of it (and so we do not not include it here).

L570-572 From the discussion in the previous section, I thought that the authors were suggesting that mesospheric QQO was related to orographic GWs.

The previous section entitled "Gravity wave interaction" noted the similarity of the pattern of GW potential energy at 10 hPa in Fig 2. of Sato et al. (2012) with the polar cap correlation plots of

Aura/MLS temperature (at 0.01 hPa (and above)) with the QQO. This was interpreted as a possible clue to the origin of the QQO without being specific about the period. The section entitled "Mechanisms for a 4-year cycle" discusses reports which include references to periods in the vicinity of a 4-year cycle.

L573-575 Did Liu et al. find any QQO in their GW potential energy at the equator. Figure 2b shows a QQO there at e.g. 81km

Liu et al. (2017) do not refer to a QQO component in their results.

L579-585 That a changing eddy diffusion is not needed anymore to explain SABER CO2 trends has nothing to do with its potential link with a QQO.

The three sentences dealing with this point have been removed.

L594 Replace the fist two dots by a comma replaced as advised

L596 Offermann et al. note in their introduction periods ranging from around 2 yrs to 11 yrs. What periods do the authors refer to in this sentence? All of them?

This sentence has been modified to focus on periods at 3.4 and 5.5 years that are closest to the QQO.

L623-624 If the polar cap also shows a QQO of similar amplitude and in phase with Davis (as you write in the next sentence, it is obvious that the Davis QQO is positively correlated with the polar cap QQO. Please, re-write.

The first point (L620-621) notes that the OH-layer equivalent temperatures of both Aura/MLS and SABER in the vicinity of the station (within 500km) agree with the Davis OH temperature QQO in amplitude, period and phase.

The following point (L622-624) notes that the Davis QQO signal is positively correlated with the Aura/MLS 0.0046 hPa level temperature field large over a large part of the polar cap and southern Indian ocean (and where it is anti-correlated in the Southern Ocean).

both points have been re-written to make this distinction clearer and the negative correlation with the NH identified as a separate point.

L631 extends vertically "at least" from the mesopause

Added text as suggested

L646 Would a significant variability in atmospheric tides be actually expected at this high polar latitudes?

No, good point, tidal variability would not be expected to be significant (tides less that 2K) but potentially their interactions with longer term oscillatory modes is responsible for the different wave patterns seen during warm and cold years of the QQO. We suggest only a potential role. See also response to L414-426.

L653-655 I do not think the authors prove this connection to be "most likely".

This point is modified to the word 'possibly' and merged with the previous point.

L669-672 Is the Davis QQO anti-correlated with the NLC boundary latitude reported by Russell et al.?

NLC data from Russell et al., 2014 are for the NH and span 2005-2011. Their fig 4 shows the NLC occurrence frequency anomaly for these years in 5° latitude bands and fig 5 the corresponding MLS and SABER [MJJA] temperatures. From our Fig 10 (see below), MLS NH summer [AMJJAS] averages show warm years in 2006 and 2009 and cold years in 2007-08 and 2010-11 when we would expect increased NLC occurrence. There is an indication of an NLC increase in 2007-08 and 2011 in their fig 4 panels h and i for the high latitude bands but the variation is small. The difficulty with the temperature/NLC correlations at high latitude are nicely explained in their summary "As latitude increases, the cloud frequency increase weakens significantly, and in high polar latitudes, the cloud frequency remains roughly constant over the 10 years examined here. This is expected because under highly saturated conditions, i.e., when temperature is far below the frost point, the cloud frequency is close to 100%, so there is no room for further increases, and the temperature control of cloud frequency no longer holds."

[Figure]

The low latitude boundary anomaly is provided in their fig 7 (reproduced right) and in summary "The results show a statistically significant increase in the number of PMCs each season in the latitude range 40°N–55°N for the 10 year period examined. Increases in cloud frequency appear to be driven by the corresponding temperature decreases over the same time period. During this time, solar activity decreased from an active to a quiet period, which might have been partially responsible for the temperature decrease over this time period." There is possible evidence of a QQO in the SABER data, but generally not enough data to be definitive.

**Figure 7.** Interannual variability of the PMC low-latitude boundary anomaly (in degrees) from its 10 year average. The anomaly is first calculated on a daily basis, and then, a seasonal average is obtained for each year. The seasonal length used here is from −10 DFS to 50 DFS when PMCs are widespread enough to cover all longitudes. The error bars are the standard error of the mean. The results show a slightly negative but statistically insignificant change in the low-latitude PMC boundary over the 10 year period used in this study.

Fig 2 caption. I think the caption for b) is not correct. If "as for a)", shouldn't it be just "correlation of the SH polar cap average", without "0.0046hPa"?

a) correlates the **0.0046hPa grid box temperature over Davis** with the MLS temperature field
b) correlates the **0.0046hPa polar cap average temperature** with the MLS temperature field both show the same plot layout of time-series and projection maps (which was the As for a) reference). We have removed the "As for a)" and specified the correlation in b) explicitly in the figure caption.

L921 What green circle?

The green circle on the SH projection map at 0.0046hPa which indicates the polar cap averaging region. It has been made a bit thicker for visibility. Figure 2 has been redone.

Figure 6. Please, indicate the maximum and minimum values in the legend of the color scale for the lower panels. Are the two color scales the same? If the answer is yes, just keep one of them. If the answer is no, use the same color scale for the four panels.

Figure 6 has been redone with the same color scale for all panels.

[Figure]

Figure 10. Please, use the same scale for both the NH and the SH residuals. Otherwise, they are not easily compared.

Figure 10 has been reformatted so the SH and NH scales are the same, but offset 3k to separate the time series.

[revised manuscript text omitted]

QQO Cold years <33%          QQO Mid years 33−67%          QQO Warm Years >67%

[Figure]

Figure S32. Composites of the ERA5 [AMJJAS] zonal wind anomaly, for cold, mid and warm years of the Davis detrended winter average QQO signal. Pressure levels are indicated on the right hand colour bar. The colour scales are in m/s. Hashed areas on the plots are significant at the 90% level.

[Figure]

Figure S4. Time series of Sea Surface Temperature (SST) anomalies for the regions marked A, B, C and D on Fig. 5 compared to the Davis OH and Aura/MLS residual temperatures. Correlation coefficients for each series are provided in the legend caption.

[Figure]

Figure S54. Davis OH winter mean residual temperatures (K) (black line; 1995-2018), and the corresponding 10 hPa (blue) and 30 hPa (yellow) standardized monthly averaged zonally averaged zonal wind (m/s) at the equator (known as the Quasi-Biennial Oscillation (QBO). QBO data were obtained from the 30 hPa and 10 hPa Singapore QBO data (https://www.geo.fu-berlin.de/en/met/ag/strat/produkte/qbo/).

[Figure]

Figure S65. Davis OH winter mean residual temperatures (K) (black line; 1995-2018), and the corresponding values of the Multivariate El Nino Southern Oscillation Index (MEIv2). The time series is bimonthly so the Jan value represents the Dec-Jan value and is centered between the months. Details and current values were obtained from NOAA ESRL (Earth System Research Laboratory) Physical Sciences Division (PSD) MEI webpage (https://www.esrl.noaa.gov/psd/data/correlation/meiv2.data).

[Figure]

Figure S7. Detrended Davis OH winter mean temperatures compared to the Annual Indian Ocean (blue line; R=-0.51) and Amundsen-Bellinghausen Sea Region (green line; R=-0.24) Sea Ice Area (Mkm² from Parkinson, 2019). Note inverted scale for sea ice area.

[Figure]

Figure S8. Wavelet spectra of CESM-WACCM Ref-C2 runs for CCMI-1 of polar cap average (65°S - 85 °S) temperature for AMJJAS. Shown are spectra for each of 3 ensemble members (columns) at three pressures (rows;  0.15 hPa, 0.5 Pa (0.005 hPa) and 0.01 Pa (0.0001 hPa)).